# LEARNING 3D-GAUSSIAN SIMULATORS FROM RGB VIDEOS

## ABSTRACT

Realistic simulation is critical for applications ranging from robotics to animation. Video generation models have emerged as a way to capture real-world physics from data, but they often face challenges in maintaining spatial consistency and object permanence, relying on memory mechanisms to compensate. As a complementary direction, we present 3DGSim, a learned 3D simulator that directly learns physical interactions from multi-view RGB videos. 3DGSim adopts MVSplat to learn a latent particle-based representation of 3D scenes, a Point Transformer for the particle dynamics, a Temporal Merging module for consistent temporal aggregation, and Gaussian Splatting to produce novel view renderings. By jointly training inverse rendering and dynamics forecasting, 3DGSim embeds physical properties into point-wise latent features. This enables the model to capture diverse behaviors, from rigid and elastic to cloth-like dynamics and boundary conditions (e.g., fixed cloth corners), while producing realistic lighting effects. We show that 3DGSim can generate physically plausible results even in out-of-distribution cases, e.g. ground removal or multi-object interactions, despite being trained only on single-body collisions.

## 1 INTRODUCTION

Simulating visually and physically realistic environments is a cornerstone for embodied intelligence. Robots must soon tackle tasks such as opening washing machines, folding laundry, or tending plants. Traditional analytical simulators demand exact geometry, poses, and material parameters, making arbitrary scene simulation impractical. An alternative is to learn models that predict future states of a scene in large-scale observations, as evidenced by the striking visual realism of 2D video generation methods (Li et al., 2022; Wu et al., 2023; NVIDIA et al., 2025). However, pure 2D approaches lack 3D structure awareness, leading to failures in occlusion handling, object permanence, and physical plausibility (Motamed et al., 2025).

3D-based representations address many of these shortcomings, as shown by recent learned particle-based simulators (Li et al., 2019; Allen et al., 2023; Zhu et al., 2024) which model a wide range of physical phenomena, from fluids and soft materials to articulated and rigid body dynamics. Yet, scaling such methods to data-rich regimes remains challenging, as most methods require privileged signals (object-level tracks, depth sensors, physics prior) or hand-crafted graph constructions.

To bridge this gap, we identify three pillars for generalizable, scalable visuo-physical simulation from videos: *(1) 3D visuo-physical reconstruction* from raw RGB observations; *(2) Imposing minimal physical biases* that can capture diverse physics; *(3) Efficient, differentiable decoding* back to image space for supervision via reconstruction loss.

Graph neural networks (GNNs) (Sanchez-Gonzalez et al., 2020; Xue et al., 2023; Whitney et al., 2023; 2024; Shi et al., 2024; Wang et al., 2024) have shown great promise in introducing relational inductive biases to handle the unstructured nature of particle sets. This has allowed GNN-based particle simulators to make major progress on all three pillars. In particular, Whitney et al. (2023) jointly train an encoder and dynamics model to learn visuo-physical pixel features from RGBD, and in the follow-up work Whitney et al. (2024) eliminate point correspondences via abstract temporal nodes or per-step models with merging. Driess et al. (2023) demonstrate end-to-end dynamics training of composable NeRF fields from raw RGB images. These advances, in combination with recent advances in feed-forward inverse rendering (Chen et al., 2024) and fast differentiable rendering of

Table 1: Overview on recently proposed particle-based simulators. While most works resort to a combination of kNN and GNNs, our work distinguishes itself by resorting to 3D Gaussian Splatting, space filling curves (SFC) for point cloud serialization, and training the inverse rendering encoder alongside a dynamics transformer.

| Method (📦: No data / code) | | Scene representation | Inverse renderer (❄: Pretrained) | Graph synthesis | Dynamics model (🔱: Uses privileged info) | Forward rendering |
|---|---|---|---|---|---|---|
| **SDF-Sim** 📦 | Rubanova et al. Rubanova et al. (2024) | Mesh | n.a. | SDF | GNN 🔱 | n.a |
| **PGNN.** | Saleh et al. Saleh et al. (2024) | Mesh | n.a. | Mesh | GNN + Attention 🔱 | n.a |
| **FIGNet** | Allen et al. Allen et al. (2022a) | Mesh faces | n.a. | BVH | GNN 🔱 | n.a |
| **Robocraft** | Shi et al. Shi et al. (2024) | Point clouds | n.a. (RGB-D) | kNN | GNN 🔱 | NeRF |
| **3DIntphys** 📦 | Xue et al. Xue et al. (2023) | Point clouds | NeRF (Point sampl.) ❄ | kNN | GNN 🔱 | NeRF |
| **VPD** 📦 | Whitney et al. Whitney et al. (2023) | Point clouds | n.a. (RGB-D + UNet) | kNN | GNN | NeRF |
| **HD-VPD** 📦 | Whitney et al. Whitney et al. (2024) | Point clouds | n.a. (RGB-D + UNet) | kNN | GNN + Transformer | NeRF |
| **DEL** 📦 | Wang et al. Wang et al. (2024) | Point clouds | NeRF (GPF) ❄ | kNN | GNN + DEM | NeRF |
| **3DGSim** | (Ours) | Gaussian splats | MVSplat | SFC | Transformer | 3DGS |

particles (Kerbl et al., 2023), encourage us to ask the question: *can we give up the inductive bias arising from locally connected graphs and still learn 3D particle-based simulators?*

To this end, we build 3DGSim, a fully end-to-end differentiable framework that embraces the power of scalable computation over hand-crafted biases. 3DGSim begins by inferring 3D visuo-physical features from raw multi-view RGB images through a feed-forward inverse renderer based on MVSplat. We then introduce a transformer-only dynamics engine, avoiding kNN-based graph construction and manually designed edge features in favor of learned spatiotemporal embeddings. Finally, a Gaussian Splatting head enables training on an image reconstruction loss from multi-view videos.

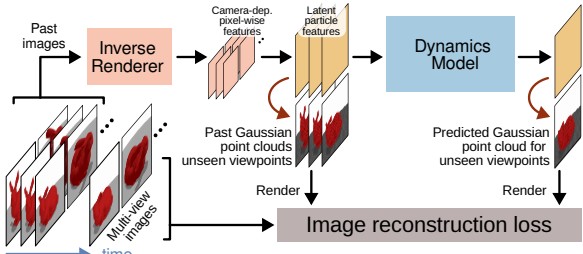

Figure 1: **3DGSim** works directly on **multi-view RGB videos** and is trained **end-to-end on next image prediction**. The **dynamics model** (transformer) operates on **particles with latent feature**. A learned mapping **transforms them into 3D Gaussian Splats** for novel view rendering.

Specifically, 3DGSim introduces the following key contributions:

- **Inverse Renderer**: Extends MVSplat with a feature extraction module fusing pixel-aligned features into a particle visuo-physical latent representation.
- **Temporal Encoding & Merging Layer**: Discards abstract temporal nodes in favor of a hierarchical module that processes an arbitrary number of timesteps.
- **Transformer-Only Dynamics Engine**: Removes graph biases and instead uses space-filling curves and learned embeddings for particle-based simulation.
- **End-to-End Differentiable Framework**: Connects inverse rendering, transformer dynamics, and Gaussian splatting-based decoding to training for next-frame image reconstruction.
- **Open Source Release**: We release the code and dataset to establish a reproducible baseline for future visuo-physical simulation research.

## 2 RELATED WORK

**Encoding and rendering scene representations** Common 3D scene representations include point clouds (particles), meshes, signed distance functions (SDFs), neural radiance fields (NeRFs) (Mildenhall et al., 2021), and 3D Gaussians (splats) (Kerbl et al., 2023). Point clouds, which approximate object surfaces, can be obtained from RGB-D sensors (Shi et al., 2024; Whitney et al., 2023; 2024) or via inverse rendering (Wang et al., 2025; Murai et al., 2024; Chen et al., 2024). Works, such as Whitney et al. (2023; 2024), use U-Net–style encoders trained jointly with the dynamics model, allowing the extracted features to be optimized for physical prediction, a strategy shown to outperform independently trained encoders (Li et al., 2022). We adopt this joint training approach using MVSplat (Chen et al., 2024), where the encoded features are initially bound to camera parameters. To unify these visuo-physical latents in a global frame, we introduce a learned feature transformation module that maps them into a consistent 3D representation. While many PBS methods render from NeRFs (Xue et al., 2023; Shi et al., 2024; Whitney et al., 2023; 2024; Wang et al., 2024; Driess et al., 2023), we instead encode visual appearance directly in the particle cloud using 3D

Gaussians. This explicit representation offers high rendering fidelity and significantly improved efficiency over NeRF-based rendering (Kerbl et al., 2023), supporting scalability.

**GNN based particle-based simulators (PBS)** Graph neural networks (GNNs) introduce relational inductive biases well-suited for modeling the unstructured nature of particle systems. Early work (Sanchez-Gonzalez et al., 2020; Li et al., 2019) demonstrated that GNN-based PBS can fit trajectories across a range of physical phenomena. However, GNNs struggle with rigid bodies, where instantaneous velocity changes require long-range message passing across the entire graph in a single step. To address this, later works incorporate mesh structures (Pfaff et al., 2021; Allen et al., 2022b) or signed distance functions (SDFs) (Rubanova et al., 2024) to enforce object-level coherence. Although effective in rigid-body settings, these methods do not generalize to deformable or fluid systems. Recent works (Saleh et al., 2024; Whitney et al., 2024) suggest adding attention layers to efficiently pass information through the graph. Wang et al. (2024) move toward greater data efficiency by incorporating physics-inspired biases such as the Material Point Method, though limiting broad applicability and requiring small simulation timesteps. To address temporal correspondence, Whitney et al. (2023) introduces abstract temporal nodes, while Whitney et al. (2024) combines GNNs with transformers to improve memory efficiency by processing and merging pairs of timesteps. However, the method is restricted to two-step horizons, as it requires training a separate model for each additional timestep. Methods based on GNNs rely on kNN to define point connectivities within a fixed radius and hand-crafted features based on object associations and distances to define graph features. Message passing and spatial pooling via furthest-point-sampling (FPS) are then used to aggregate information for dynamics prediction. However, kNN and distance computations are expensive and take up 54% of the forward time (Wu et al., 2024b), which limits scalability and prevents real-time forecasting. In contrast, we follow the design of PTv3 (Wu et al., 2024b). In 3DGSim, we trade off exact KNN neighborhood computation with space-filling curve–based ordering of particles and use sparse convolutions to encode relative positions, avoiding distance calculations. To enable the processing of temporal point clouds, we propose Temporal Merging with Grid Pooling to construct a hierarchical spatiotemporal, UNet-style Point Transformer for dynamics prediction.

**Analytical particle simulators as physical prior** Our work differs in purpose from applications which use Gaussian Splatting particles and analytical PBS as physical prior (e.g. off-the-shelf differentiable MPM simulator) to accomplish a series of tasks such as tracking (Luiten et al., 2024; Keetha et al., 2024; Zhang et al., 2024a; Abou-Chakra et al., 2024), dynamic scene reconstruction (Wu et al., 2024a; Huang et al., 2023; Yu et al., 2023), or animation (Xie et al., 2023; Zhang et al., 2024b; Lin et al., 2025). While analytical PBS can be used for parameter identification (Abou-Chakra et al., 2024), they are tailored to specific simulation scenarios. For a detailed comparison, refer to the supplementary material (see Appendix C.2).

## 3 PRELIMINARIES

3DGSim is build atop several prior works, namely: 3D-Gaussian splatting which enables fast rendering, MVSplat which yields 3D Gaussian point clouds from multi-view images, and PTv3 which enables efficient neural processing of 3D point clouds.

**Gaussian Splatting** 3D Gaussian splatting (3DGS) (Kerbl et al., 2023) is an effective framework for multi-view 3D image reconstruction, representation, and fast image rendering and has gained rapid popularity due to its support for rapid inference, high fidelity, and editability of scenes. Gaussian splatting uses a collection of 3D Gaussian primitives, each parameterized by

$$g_i = (p_i, c_i, r_i, s_i, \sigma_i), \tag{1}$$

with the Gaussian's mean $p_i$ (particle position), its rotation $r_i$, spherical harmonics $c_i$ (defines coloring), scale $s_i$, and opacity $\sigma_i$. To render novel views, these primitives are projected onto a 2D image plane using differential tile-based rasterization. The color value at pixel $\mathbf{p}$ is calculated via alpha-blend rendering: $I(\mathbf{p}) = \sum_{i=1}^{N} \alpha_i c_i \prod_{j=1}^{i-1} (1 - \alpha_j)$ where $\alpha_i = \sigma_i e^{-\frac{1}{2}(\mathbf{p}-p_i)^\top \Sigma_i^{-1}(\mathbf{p}-p_i)}$ is the 2D density, $I$ is the rendered image, $N$ is the number of primitives in the image and $\Sigma_i$ is the covariance matrix given by $\Sigma_i = r_i s_i r_i^\top$ for improved computational stability.

**MVSplat: Multi-view feed-forward 3D reconstruction**
MVSplat deploys a feed-forward network $f_\phi$ with parameters $\phi$ that maps $M$ images $\mathcal{I} = \{I^m\}_{i=m}^M$ with $I^m \in \mathbb{R}^{(H \times W \times 3)}$ to a set of pixel-aligned 3D Gaussian primitives (Fig. 2)

$$f_\phi : \{(I^m, P^m)\}_{m=1}^M \mapsto \{g_i\}_{i=1}^{M \times H \times W}.$$

At each time step, MVSplat localizes Gaussian centers using a cost volume representation through plane-sweeping and cross-view feature similarities. To do so, it requires the corresponding camera projection matrices $\mathcal{P} = \{P^m\}_{m=1}^M$ that are calculated as $P^m = K^m[R^m|t^m]$ via the camera intrinsics $K^m$, rotation $R^m$, and translation $t^m$.

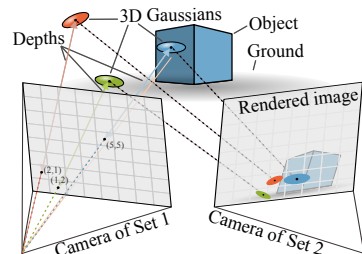

Figure 2: MVSplat uses a cost volume with plane sweeping to regress pixel-wise 3D Gaussians, which are unprojected to world frame using camera parameters.

# 4 3DGSIM

3DGSim is a fully differentiable pipeline that, given $T$ past multi-view RGB frames, reconstructs 3D particles with latent features, simulates their motion, and renders the next frames. It consists of three jointly trained modules (Fig.1): (i) an *encoder* that maps multi-view RGB images to 3D particles, (ii) a *dynamics model* that simulates the motion of these particles through time, and (iii) a *renderer* that yields images by first mapping the particles to Gaussian splats.

## 4.1 STATE REPRESENTATION

To simulate physical scenes from vision, we require a state representation that is both expressive enough to capture fine-grained 3D and physical properties, and compact enough to enable efficient learning and prediction. Although an explicit 3DGS representation $g_i(t_k)$ offers geometric and visual completeness, it is insufficient for dynamics modeling. Instead, we distill the state of each particle into a more compact representation:

$$\tilde{g}_i(t_k) = \big(p_i(t_k),\ f_i(t_k)\big) \tag{2}$$

where $t_k$ denotes the $k$-th timestep, $p_i$ the position and $f_i \in \mathbb{R}^d$ the visuo-physical latent particle feature, encoding shape, appearance, and dynamic properties. Unless otherwise stated, we omit the timestep $t_k$ and the particle index $i$ when the statement applies to all timesteps or particles, respectively.

**Optional: Masking and Freezing of Particles** At each timestep $t_k$, the encoder yields pixel-aligned features for each input image. As an optional step, one can apply a foreground mask to discard particles likely belonging to the static background, retaining a reduced set of $N_k$ particles per time step (Fig. 2). Additionally, as originally suggested by Whitney et al. (2023), static particles can optionally be "frozen", i.e. act as input to the dynamics model but are excluded from position updates. These optional strategies improve efficiency without being necessary for successful training, as shown in Section 5 and Appendix B.

**Invariant and dynamic feature decomposition** We decompose each particle's visuo-physical feature into an invariant and a dynamic part as shown in Fig. 3, writing

$$f_i = f_i^{\text{inv}} \oplus f_i^{\text{dyn}},$$

where $\oplus$ denotes concatenation. The dynamics model updates only $f_i^{\text{dyn}}$, while leaving $f_i^{\text{inv}}$ unchanged. For clarity, we will refer to the dynamics update as "updating $f_i$", though only the dynamic component $f_i^{\text{dyn}}$ is altered.

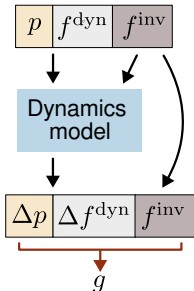

Figure 3: Position $p$ and dynamic features $f^{\text{dyn}}$ are updated while $f^{\text{inv}}$ remain constant.

## 4.2 VIEW-INDEPENDENT INVERSE RENDERER

In MVSplat, pixel-aligned features $\hat{f}_i'$ are tied to the specific camera view from which they were extracted. While Gaussian primitives (e.g. depth, scale, rotation, harmonics) can be directly unprojected or transformed into the world frame using camera parameters, latent features remain bound to the camera-centric frame. Since dynamics predictions are invariant to the observer's viewpoint, such a dependence on view-dependent encodings hampers generalization.

To overcome this, 3DGSim introduces a feature encoding network that maps pixel-aligned features $\hat{f}_i'$ into view-independent latent representations $f_i$. The encoder employs FiLM conditioning (Perez et al., 2017) on pixel depth, pixel shift, density, and ray geometry (parameterized via Plücker coordinates (Plücker, 1868-1869)) to infer spatially consistent 3D features. As a result, the inverse rendering module produces canonically anchored particle states, providing a unified representation for downstream dynamics learning. Further architectural details are described in Appendix A.1.

## 4.3 DYNAMICS MODEL

At the core of our method is the dynamics model, a transformer architecture operating on particle sets in space and time. The dynamics model receives as input $T$ past particle sets,

$$\left\{ \{\tilde{g}_i(t_k)\}_{i=1}^{N_k} \right\}_{k=1}^T, \quad \text{where } \tilde{g}_i(t_k) = \left( p_i(t_k), f_i^{\text{inv}}(t_k), f_i^{\text{dyn}}(t_k) \right), \tag{3}$$

and predicts the updated dynamic features at the next timestep

$$\Delta p(t_T), \Delta f^{\text{dyn}}(t_T) = \text{Dynamics Model} \left( \{\{\tilde{g}_i(t_k)\}_{i=1}^{N_k}\}_{k=1}^T \right), \tag{4}$$

such that $p_i(t_{T+1}) = p_i(t_T) + \Delta p_i(t_T)$ and $f_i^{\text{dyn}}(t_{T+1}) = f_i^{\text{dyn}}(t_T) + \Delta f_i^{\text{dyn}}(t_T)$. As these point clouds are unstructured and potentially vary in size at each time step due to masking, a fundamental challenge arises: *How can a network efficiently propagate the embedded physics information both spatially and temporally?*

We tackle this question by building on PTv3 Wu et al. (2024b), which has recently achieved state-of-the-art performance in representation learning for unstructured point clouds Wu et al. (2025). As discussed in Appendix A.2, PTv3 operates by serializing the input point cloud and applying patch-wise attention. However, the original design of PTv3 is limited to point clouds that do not exhibit temporal variation. In this section, we extend PTv3 to predict dynamics from *temporally evolving point clouds*. First, we extend serialization to equip point cloud encodings with a timestamp. Then, we equip features with temporal embeddings that allow attention to distinguish timestamps. Lastly, we use the timestamps to merge neighboring latent particle sets, enabling PTv3's patch-wise attention blocks to aggregate information across time.

**Temporally serialized point cloud (t-SPC)** To enable spatio-temporal reasoning over multiple timesteps, we extend PTv3's point serialization scheme by encoding both spatial and temporal structure into a single key. Specifically, for each particle $i$ at timestep $t_k$ in batch $b$, we define a 64-bit serialization code:

$$\tilde{s}_i(t_k, b) = \left[ \underbrace{b}_{(64 \text{ - } \tau \text{ - } \kappa)\,\text{Bits}} \mid \underbrace{s_{t_k}}_{\tau\,\text{Bits}} \mid \underbrace{s_i}_{\kappa\,\text{Bits}} \right] \tag{5}$$

Here, $s_{t_k}$ is the temporal code and $s_i$ is a spatial code obtained by projecting $p_i$ onto a space-filling curve (SFC). We set $\kappa = 48$ and allocate $\tau = \log_2(T)$ bits for time. With 16 bits per dimension and a grid resolution of $G = 0.004\,\text{m}$, the spatial encoding spans up to $216\,\text{m}$ per axis.

**Temporal encoding** As shown in Fig. 6, before merging t-SPCs across timesteps, we inject a learned, timestep-specific positional encoding $E_{t_k}$ as

$$f_i(t_k) \leftarrow f_i(t_k) + E_{t_k}. \tag{6}$$

This temporal encoding ensures that the attention mechanism can distinguish points across different temporal instances, enabling the model to reason about dynamics over time. Similar positional encoding methods have previously been applied in transformer architectures to differentiate positions within sequences (Vaswani et al., 2017).

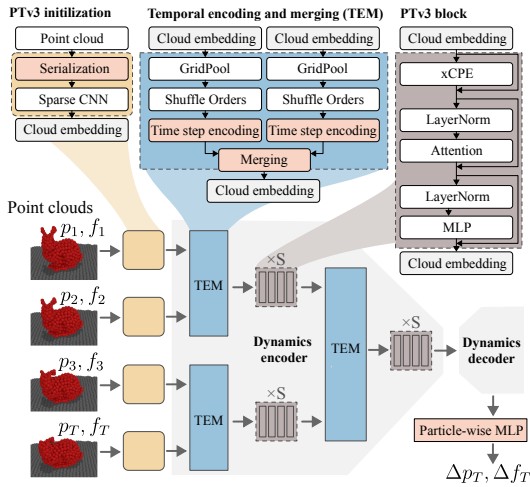

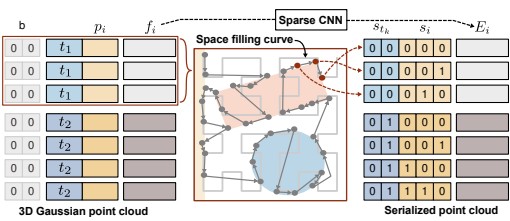

Figure 5: Spatio-temporal point cloud serialization.

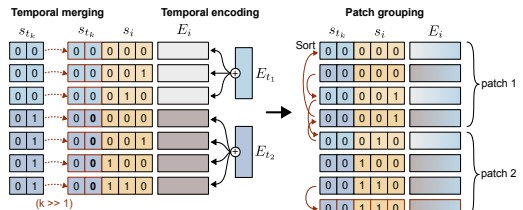

Figure 4: **The dynamics model** encodes the time step into each embedding and merges embeddings from adjacent timesteps. The TEM and PTv3 blocks are applied repeatedly until all embeddings are merged. Our extensions to PTv3 are highlighted in red.

Figure 6: Temporal merging and embedding followed by patch grouping for applying patch-wise attention.

**Temporal merging** Unlike PTv3, which restricts attention exclusively to patches composed of points from the *same time step*, our method enables a wider receptive field across time. To do so, we propose *temporal merging* which applies a one-bit right shift to the temporal codes $s_{t_k}$:

$$\text{Merge}(\tilde{s}_i) = [b \mid (s_{t_k} \gg 1) \mid s_i]. \tag{7}$$

For instance, points from time steps $s_{t_1} = 0$ and $s_{t_2} = 1$ are merged by shifting their codes, so they both become 0, as depicted in Fig. 6. By grouping points from separate time steps into a single patch, the attention module can model relationships across time.

Importantly, while Whitney et al. (2024) deploy a dedicated transformer module for each time step, our proposition of temporal merging enables the reuse of the same attention block across time steps, which significantly reduces memory consumption and promotes knowledge transfer.

**Patch-wise attention and particle-wise MLP** After each temporal encoding and merging (TEM) block, the cloud embeddings are processed by PTv3's patch-wise attention block. First, the embeddings are equipped with a position encoding via a sparseCNN with skip connection (xCPE in Fig. 4). Then, the embeddings are fed to a patch-wise attention layer. Finally, at the end of the dynamics model, each particle alongside its embedding is mapped by a particle-wise MLP to $\Delta p_T$ and $\Delta f_T$.

### 4.4 RENDERING FEATURES FOR THE IMAGE RECONSTRUCTION LOSS

To render images with 3DGS, particle states $\tilde{g}_i = (p_i, f_i)$ are transformed into Gaussian splat parameters $g_i$ via a learned head, materialized only at the final stage to supervise the training with image reconstruction.

3DGSim is trained solely on an image reconstruction loss $\mathcal{L}$. This loss is computed from rasterized multi-view images, generated based on both the encoder predictions of past point clouds $\{\{g_i(t_k)\}_{i=1}^{N_k}\}_{k=0}^{T}$ and the simulated future point cloud trajectory $\{\{g_i(t_k)\}_{i=1}^{N_k}\}_{k=T+1}^{T+T'}$. Specifically, the loss reads

$$\mathcal{L} = (1-\lambda)\frac{1}{T}\sum_{k=0}^{T}\mathcal{L}_k + \lambda\sum_{k=T+1}^{T+T'}\gamma^{k-T-1}\mathcal{L}_k \quad \text{and} \quad \mathcal{L}_k = \mathcal{L}_2(I_k^{\text{gt}}, I_k) + \beta\,\mathcal{L}_{\text{LPIPS}}(I_k^{\text{gt}}, I_k), \tag{8}$$

with $\lambda = 0.5$, temporal decay factor $\gamma = 0.87$, $T \in \{2, 4\}$ and $T' = 12$. The per-frame reconstruction loss $\mathcal{L}_k$ measures the discrepancy between ground-truth ($I_k^{\text{gt}}$) and predicted ($I_k$) multi-view images using a weighted combination of pixel-wise $\ell_2$ and LPIPS Zhang et al. (2018) terms with hyperparameter $\beta = 0.05$.

## 5 EXPERIMENTS

In what follows, we train 3DGSim on different datasets and test the model's ability to generalize.

**Model setup** Unless stated otherwise, the following training and parameter settings serve as defaults in the experiments. The state consists of dynamic $f^{\text{dyn}}$ and invariant features $f^{\text{inv}}$ of size $(32, 32)$ for the implicit- and $(n_f, 16)$ for the explicit 3D Gaussian particle representation. In the explicit representation, $f^{\text{dyn}}$ corresponds to explicit Gaussian primitives of size $n_f$ which are directly used for rendering. The inverse rendering encoder follows MVSplat, reducing candidate depths from 128 to 64 due to smaller scene distances. Default near-far depth ranges are $[0.2, 4]$ for rigid bodies and $[1.5, 8]$ for the other datasets, as the scene has a larger scale. The dynamics transformer defaults to PTv3 with a 5-stage encoder (block depths $[2, 2, 2, 6, 2]$) and a 4-stage decoder $([2, 2, 2, 2])$. Grid pooling and temporal merging strides default to $[1, 4, 2, 2, 2]$ and $[1, 2, 2, 2, 2]$, respectively, with grid size $G{=}0.004$ m. Attention blocks use patches of size 1024, encoder feature dimensions $[32, 64, 128, 256, 512]$, decoder dimensions $[64, 128, 256]$, encoder heads $[2, 4, 8, 16, 32]$, and decoder heads $[4, 4, 8, 16]$. For the camera setup, we select 4 uniformly distributed views at random and an additional 5 target cameras from the remaining cameras (out of 12 total) to compute the reconstruction loss.

**Training** Our models are trained with AdamW for $\sim$120,000 steps using a cosine annealing warm-up and a learning rate of $2 \times 10^{-4}$, with batch sizes of 6 and 4 for 2-step and 4-step states, respectively. To optimize memory and speed, we use gradient checkpointing and flash attention v2 (Dao, 2024). Training is performed on a single H100 GPU and typically takes around six days. During training, no special treatment is applied to the ground plane. It is treated the same as the rest of the scene, consisting of a set of particles which the model has learned to update accordingly.

**Datasets** To evaluate 3DGSim's robustness in learning dynamics from videos, we introduce three challenging datasets: rigid body, elastic, and cloth.

The rigid body dataset consists of 1,000 simulated trajectories from the MOVI dataset, involving six rigid objects (turtle, sonny school bus, squirrel, basket, lacing sheep, and turboprop airplane) from the GSO dataset (Downs et al., 2022). Each trajectory spans 32 frames at 12 FPS, providing controlled dynamics characteristic of rigid body motion. The elastic dataset, aimed at capturing plastic deformable object dynamics, includes six objects (dragon, duck, kawaii demon, pig, spot, and worm) simulated using the Genesis MPM elastoplastic simulator (Authors, 2024). Each object undergoes deformation upon collision with a circular gray ground, offering scenarios of complex elastic behavior. The cloth dataset includes the same set of objects as the elastic dataset. Here, the cloth is fixed at four corners, posing the challenge to infer implicit constraints and modeling dynamic cloth-like deformations.

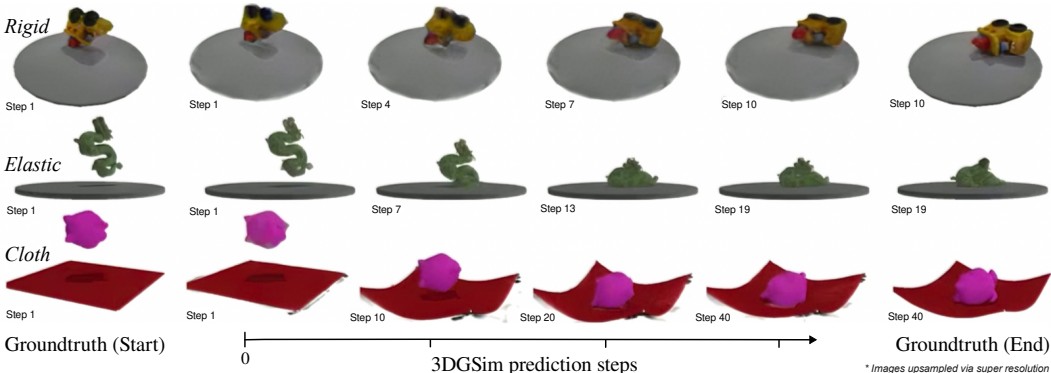

Figure 7: Qualitative examples of 3DGSim's dynamic predictions. After training on less than 6 minutes of video per object across 6 objects, 3DGSim accurately predicts motion of elasto-plastic deformations, rigid bodies, cloth. The first and last column represent the initial and last frame of the "true" simulated motion. The in-between columns/frames are predictions of 3DGSim.

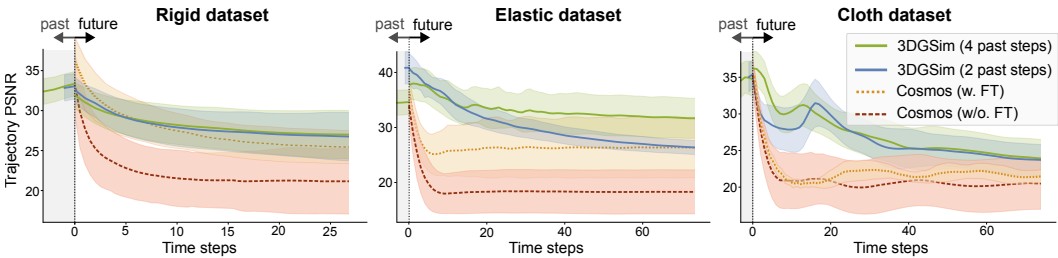

Figure 8: Trajectory PSNR of 3DGSim, Cosmos and CosmosFT is shown for both past and future predictions. The Cosmos models are conditioned on past frames and appropriate language prompts.

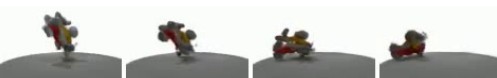

Figure 9: 3DGSim's prediction of a rigid plane captures shadows by altering ground particle appearance.

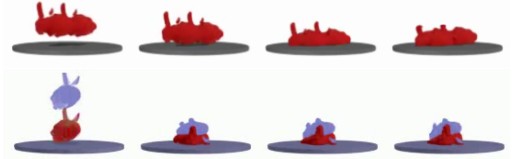

Figure 10: Although not trained on this specific elastic object or multiple objects, 3DGSim predicts physically plausible deformations.

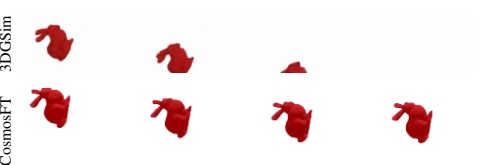

Figure 11: When the ground is removed, 3DGSim predicts the freefall, while CosmosFT hallucinates a levitating object at ground level.

Figure 12: Comparison of 3DGSim to Cosmos. *Explicit* models use 3DGS parameters and a static latent feature as inputs to the dynamics model, while *latent* models use only latent features mapped to Gaussians after dynamics. The †-model uses only 6 camera views (3 input + 3 reconstruction) instead of 12 (4+5); the ‡-model omits segmentation masks for static elements. Metrics *future* and *past* are means over all timesteps. "4-12" means 4 past steps predicting 12 future steps.

| Dataset | Model | PSNR (future) ↑ | PSNR (past) ↑ | SSIM ↑ | LPIPS ↓ |
|---|---|---|---|---|---|
| Rigid | 3DGSim 4-12 latent | **28.28 ± 2.52** | 32.93 ± 1.56 | **0.90 ± 0.03** | **0.09 ± 0.03** |
| | 3DGSim 2-12 latent | **28.08 ± 2.46** | 33.00 ± 1.62 | **0.90 ± 0.03** | **0.09 ± 0.03** |
| | 3DGSim 4-12 explicit | 27.88 ± 2.43 | 32.77 ± 1.57 | 0.90 ± 0.03 | 0.09 ± 0.03 |
| | 3DGSim 2-12 explicit | 27.07 ± 2.27 | 32.67 ± 1.65 | 0.90 ± 0.03 | 0.09 ± 0.03 |
| | CosmosFT | 26.44 ± 2.26 | – | 0.68 ± 0.05 | 0.10 ± 0.03 |
| | Cosmos | 22.35 ± 3.82 | – | 0.83 ± 0.08 | 0.24 ± 0.08 |
| Elastic | 3DGSim 4-12 latent | **33.15 ± 3.51** | 34.55 ± 2.26 | **0.97 ± 0.02** | **0.02 ± 0.01** |
| | 3DGSim 2-12 latent | **32.05 ± 3.48** | 35.99 ± 1.88 | 0.96 ± 0.02 | 0.03 ± 0.02 |
| | 3DGSim 2-12 explicit | 29.92 ± 1.72 | 40.85 ± 2.94 | 0.96 ± 0.02 | 0.03 ± 0.02 |
| | 3DGSim 4-12 explicit | 29.69 ± 1.75 | 40.16 ± 3.07 | 0.97 ± 0.02 | 0.02 ± 0.01 |
| | 3DGSim 4-12 latent † | **31.60 ± 3.09** | 32.55 ± 2.12 | **0.97 ± 0.02** | **0.02 ± 0.01** |
| | 3DGSim 4-12 latent ‡ | **32.66 ± 3.43** | 34.45 ± 2.44 | 0.96 ± 0.02 | 0.03 ± 0.02 |
| | CosmosFT | 26.50 ± 5.21 | – | 0.82 ± 0.02 | 0.07 ± 0.03 |
| | Cosmos | 18.87 ± 3.99 | – | 0.79 ± 0.08 | 0.23 ± 0.08 |
| Cloth | 3DGSim 4-8 latent | **26.98 ± 2.63** | 34.81 ± 2.28 | **0.89 ± 0.03** | **0.08 ± 0.03** |
| | 3DGSim 2-8 latent | **26.25 ± 2.38** | 35.22 ± 1.97 | **0.88 ± 0.03** | **0.08 ± 0.02** |
| | 3DGSim 4-8 explicit | 23.72 ± 1.52 | 39.75 ± 2.32 | 0.89 ± 0.03 | 0.08 ± 0.03 |
| | 3DGSim 2-8 explicit | 17.97 ± 2.02 | 35.47 ± 1.68 | 0.88 ± 0.03 | 0.08 ± 0.02 |
| | CosmosFT | 22.49 ± 0.99 | – | 0.73 ± 0.03 | 0.14 ± 0.04 |
| | Cosmos | 21.10 ± 3.56 | – | 0.86 ± 0.06 | 0.19 ± 0.06 |

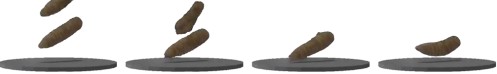

Figure 13: CosmosFT merges distinct worms into one before ground contact, even on in-distribution cases.

Both elastic and cloth datasets include 200 trajectories per object, simulated with a 0.001 time step and 20 substeps. Each two second sequence is recorded at 42 FPS resulting in 84 frames per trajectory and less than 6 minutes of footage per object.

**Benchmarking** Existing 3D baselines do not allow direct comparison without substantial reimplementation. Key methods – VPD, HD-VPD, DEL, and 3D-IntPhys (Whitney et al., 2023; 2024; Wang et al., 2024; Xue et al., 2023) – *lack public code and data*, also unavailable upon contacting authors. Without published datasets, any reimplementation would lack verifiability, limiting reproducibility and fair evaluation. To address this, we will release our code and datasets. DPI-Net and VGPLDP (Li et al., 2019; 2020) are open-source but rely on ground-truth particle trajectories and require major adaptation to fit our setting. For 2D baselines, we provide quantitative comparison to Cosmos (NVIDIA et al., 2025). Note that Cosmos differs from our multi-view setup as it is pretrained on multiple past frames from a single view. For fair comparison, we evaluated both the base model and a LoRA-finetuned variant of Cosmos-Predict2 (CosmosFT) trained for 6,000 iterations on our data set using recommended parameters. The Cosmos models are conditioned on the prompts detailed in Table S7). For evaluation, 12% of trajectories are chosen at random and held out from each dataset, and we report each model's PSNR, LPIPS and SSIM.

## 5.1 TRAJECTORY SIMULATION

3DGSim achieves competitive long-horizon simulation accuracy; up to 80 steps; compared with state-of-the-art baselines such as Cosmos-1.0-Autoregressive-5B-Video2World and a LoRA-finetuned Cosmos-Predict2 (NVIDIA et al., 2025). Performance curves are shown in Fig. 8. Ablation studies

(Tab. 12) reveal that keeping 3DGS primitives explicit in the representation yields similar short-term performance but generalizes poorly, especially with fewer cameras (see Appendix B). By contrast, using a latent implicit representation leads to more robust generalization.

## 5.2 ABLATION

We use the elastic dataset to ablate the contribution of all key components of 3DGSim.

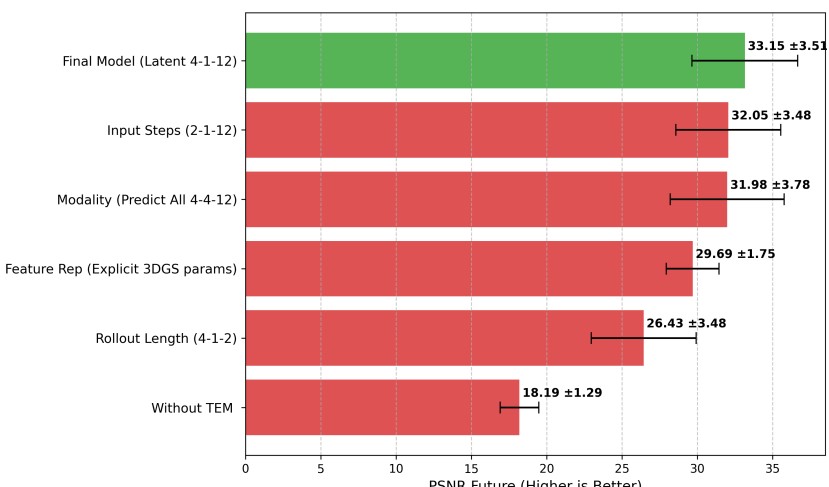

Figure 14: Ablation Study: The table reports the performance of the Final Model (3DGSim) alongside variants where individual elements, such as Temporal Encoding/Merging, rollout length, feature representation, modality, and input steps, are removed or modified. The notation *4-1-12* indicates: a 4-step input window, predicting deltas only for the last step, and training with a 12-step future rollout.

**TEM is Critical:** Removing the Temporal Encoding and Merging (TEM) module causes a massive performance collapse (dropping to 18 PSNR), indicating that simple spatial attention is insufficient.

**Latent > Explicit:** The latent representation (Final Model) outperforms the Explicit representation (29.69 PSNR). By keeping the representation abstract, the model can embed physical properties (like mass or friction) into the latent space instead of overfitting to the visual aspects.

**Rollout Length:** Training on longer horizons (12 steps) significantly boosts performance compared to short horizons (2 steps, 26.43 PSNR), likely due to better long-term stability learning.

## 5.3 SCENE EDITING AND MODEL GENERALIZATION

With its explicit 3D state, 3DGSim supports direct scene editing, providing a natural testbed for generalization. When the ground is raised or removed, conditions never seen during training, the model continues to generate stable, physically consistent rollouts (Fig. 11). This suggests a robust grasp of underlying dynamics that extends beyond the training distribution.

We further test generalization by duplicating objects and running long-horizon simulations (Fig. 10), Appendix D.1). Although trained only on single object–ground collisions, 3DGSim accurately captures realistic multi-body interactions, with objects retaining integrity rather than collapsing into chaotic overlaps. Beyond interactions, it even models emergent properties such as shadows (Fig. 9), indicating a holistic understanding of lighting and geometry alongside physics.

In contrast, CosmosFT struggles under similar 2D-edits. When the ground is removed, objects often remain suspended (Fig. 11), and when multiple objects are introduced, they morph into a single mass before contact (Fig. 13). These hallucinations reflect the limits of 2D image-based reasoning, underscoring the advantages of an explicit 3D representation for robust and interpretable generalization. Further examples are shown in the supplementary.

## 5.4 SIMULATION SPEED

Simulation speed is critical for robotics applications. Traditional simulators (FEM, MPM, PBD) typically employ small integration timesteps. Learned approaches enable larger timesteps, allowing 3DGSim to simulate elastic cloth at 42 FPS and rigid dynamics at 12 FPS, with inference speeds of ~16 FPS (4 past steps) and ~20.1 FPS (2 past steps), using under 20 GB VRAM on an H100 GPU and achieving *near real-time speeds* as illustrated in Fig. 15.

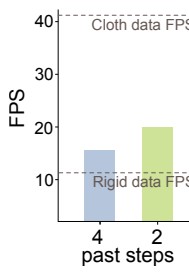

Figure 15: Prediction speed of 3DGSim versus simulation FPS.

## 6 DISCUSSION

We introduced 3DGSim, a fully differentiable 3D Gaussian simulator that learns directly from multi-view RGB video. 3DGSim integrates inverse rendering, dynamics prediction, and novel-view video synthesis within a single end-to-end learnable system. Given that 3DGSim pioneers an unexplored direction for 3D particle-based simulation, future work will explore action conditioning, a natural next step that provides essential supervision signals for forecasting. This will also enable large-scale validation on real-world multi-view datasets, which are currently unavailable for passive phenomena. Our dependence on multi-view inputs could be further mitigated by recent advances in monocular inverse rendering (Wang et al., 2025; Murai et al., 2024). Additionally, while occlusions are not explicitly modeled, they are partially addressed by the dynamics module and may be further improved through point completion techniques.

**Spatial Causality** In 3DGSim, interactions are restricted to those between spatially grounded particles, which ensures that the simulation adheres to realistic physical dynamics. This contrasts with 2D pixel-based video generation models, where apparent dynamics often emerge from the generative flexibility of image-space synthesis. The 3D formulation thus brings advantages such as spatial consistency, object permanence, and robustness to out-of-distribution inputs, as exemplified by our generalization tests. However, it introduces certain compromises: while 2D predictors can effortlessly repurpose pixels to synthesize novel content, e.g., fabricating unseen objects from generative priors, 3D particle models inherit stricter structural constraints, making it difficult to dynamically create or destroy particles in a learnable manner. This highlights a tradeoff between the stability and interpretability provided by 3D spatial causality and the generative freedom unlocked by 2D video models.

**Toward Vision-Language Simulation** Integrating language embeddings offers a promising avenue for enriching particle-based simulations. Terms such as "liquid" or "mirror" provide informative priors about object properties, enabling more structured and semantically aware predictions. We envision 3DGSim as a step toward scalable simulators that can learn physical interactions from both visual and textual modalities, ultimately supporting a more nuanced robotic understanding of complex real-world dynamics.

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

# A   3DGSIM ADDITIONAL DETAILS

## A.1   UNPROJECTING PIXEL-ALIGNED FEATURES VIA FiLM CONDITIONING

To transform the view-dependent pixel-aligned features $\hat{f}'_i$ into a consistent 3D representation, we use a multilayer perceptron (MLP) with Feature-wise Linear Modulation (FiLM). FiLM conditioning enables the MLP to adapt its processing of $\hat{f}'_i$ based on geometric context, such as camera viewpoint and depth.

Specifically, FiLM computes a scale and bias using a conditioning network $\gamma$ that takes as input a geometric context vector $\mathbf{x}_i$—which includes depth, density, and pixel shift—as well as the Plücker ray encoding $\mathbf{r}_i = [\mathbf{o}_i \times \mathbf{d}_i \mid \mathbf{d}_i,]$, where $\mathbf{o}_i$ and $\mathbf{d}_i$ denote the origin and direction of the viewing ray:

$$\text{scale}_i,\ \text{bias}_i = \gamma(\mathbf{x}_i, \mathbf{r}_i). \tag{S1}$$

These parameters modulate the activations of the MLP processing $\hat{f}'_i$ through FiLM layers:

$$f_i = \text{MLP}(\hat{f}'_i;\ \text{scale}_i,\ \text{bias}_i), \quad \text{FiLM}(h) = \text{scale}_i \odot h + \text{bias}_i, \tag{S2}$$

where $h$ denotes a hidden activation and $\odot$ is element-wise multiplication.

This setup allows the network to unproject image-aligned features into canonical 3D space while respecting scene geometry and view direction.

## A.2   PTv3: SCALABLE POINT CLOUD TRANSFORMATIONS

**Point cloud serialization**   At the core of PTv3 lies "point cloud serialization", an algorithm that transforms an unstructured point cloud into ordered points. This process begins by discretizing 3D space into a uniform grid of points. As illustrated in Fig. 5, these point are then connected using a space-filling curve – a path that traverses each grid point exactly once while preserving spatial proximity as much as possible. Each point $p_i$ is assigned an integer code $s_i$, representing its position within a space-filling curve, via the mapping

$$s_i = \phi^{-1}(\lfloor p_i/G \rfloor) \tag{S3}$$

with $\phi^{-1} : \mathcal{Z}^N \mapsto \mathcal{Z}$ and grid size $G \in \mathbb{R}$. The points in the clouds are then ordered by their respective code $s_i$, yielding a serialized point cloud (SPC)

$$S_i(t_k) = \{(s_i, \tilde{g}_i)\}_{i=1}^{N_k}. \tag{S4}$$

While this approach may not preserve local connectivity as precisely as kNN groupings, Wu et al. (2024b) emphasizes that the slight loss in spatial precision is outweighed by a significant gain in computational efficiency. To obtain diverse spatial connections between points, PTv3 shuffles between four different space filling curve patterns to obtain SPCs from which patches are computed and varies the patch computation through integer dilation.

**Patch grouping**   PTv3 partitions the SPC $S_i(t_k)$ into equally sized patches and applies self-attention within each patch. To ensure divisibility, patches that do not align with the specified size are padded by borrowing points from neighboring patches.

**Conditional embeddings and patch attention**   Besides the computational efficiency of SPC over kNN, its main advantage lies in the compatibility with standard dot-product attention mechanisms. To understand this, we first examine how the standard PTv3 architecture computes particle-wise predictions from a single point cloud at a single time step. The process begins by extracting particle-wise embeddings $E_i$ for each serialized point $(s_i, \tilde{g}_i)$ using a sparse convolutional neural network (CNN). Next, the embedded SPCs are progressively down sampled via grid pooling before being grouped and shuffled into patch pairs. Conditional positional embeddings (xCPE) are then added to the embeddings, followed by layer normalization and a patch-wise attention layer predicting the change in the embeddings $\Delta E_i$. The pooling, patch shuffling, and attention blocks are arranged in a U-net Ronneberger et al. (2015) like architecture that first reduces the size of the SPCs in an encoding step and then mirrors this architecture. In 3DGSim, the final layer of the dynamics model predicts the change in the particle positions $\Delta p_i$ and the change in their features $\Delta f_i$.

### A.3 ARCHITECTURES

This section describes additional implementation details that were not explicitly mentioned in the main paper.

**Feature Encoding Network**    In 4.2 we describe the feature encoding network that transforms pixel-aligned features into view-independent the latent features. To regress the FiLM conditioning we use a 2-layer CNN with GELU activation, kernel of size 3 and channel dimensions $(10, 20)$. The first dimension also matches the size of the conditioning vector.

**Particle Wise MLP**    At the end of the dynamics model, the embedding of each particle is mapped back to the particle latent features. For that we use an MLP with shapes $(128, 128)$ and GELU activation between each layer.

## B    ABLATIONS

For the ablation studies of 3DGSim, only the elastic dataset is used. The default configuration uses the latent representation, 4-step past conditioning, 12-step future rollouts, 4 input views, 5 target views, and a total of 12 cameras for training, unless otherwise specified. Any deviations from these parameters are explicitly stated or made clear within the context of the respective ablation.

This analysis provides detailed insights into design and strategic choices that inform future improvements of our approach.

### B.1    ROLLOUT LENGTH.

| Rollout Length | PSNR Future | PSNR Past |
|---|---|---|
| 2 steps | $26.43 \pm 3.48$ | $35.70 \pm 2.51$ |
| 4 steps | $28.83 \pm 4.96$ | $35.25 \pm 2.53$ |
| 8 steps | $30.64 \pm 3.23$ | $33.86 \pm 2.17$ |
| **12 steps** | $\mathbf{33.15 \pm 3.51}$ | $\mathbf{34.55 \pm 2.26}$ |

Table S1: Rollout Length

We evaluate the influence of prediction rollout length during training (2, 4, 8, and 12 steps). Consistent with expectations, results improve significantly as the rollout length increases, reaching a peak performance at 12 steps with a PSNR Future of $33.15 \pm 3.51$. Extending the number of rollout steps enhances the model's predictive capability but leads to significant memory requirements. Employing regularization methods such as random-walk noise injection or diffusion techniques or even other modality (see later) can help reduce the required rollout steps.

### B.2    CAMERA SETUP.

| Setup | PSNR Future | PSNR Past |
|---|---|---|
| Explicit 3 views out of 6 | $21.02 \pm 1.78$ | $16.86 \pm 0.83$ |
| **Latent 3 views out of 6** | $\mathbf{31.60 \pm 3.09}$ | $\mathbf{32.55 \pm 2.12}$ |
| **Latent 4 views out of 12** | $\mathbf{33.15 \pm 3.51}$ | $\mathbf{34.55 \pm 2.26}$ |

Table S2: Camera Setup (85k steps)

To approximate a realistic scenario suitable for real-world deployment, we investigate performance with reduced camera setups. Interestingly, the latent representation models achieve robust performance even when trained with 3 views out of 6 total cameras (PSNR 33.15), whereas explicit representation models degrade significantly (PSNR drops to 21.02) due to convergence of the encoder to poor local minima. This local minimum manifests as camera-specific overfitting, where the

model erroneously predicts particle arrangements forming planar, screen-aligned shapes. While this artificially reduces the training loss of target viewpoints, it undermines the true representation quality and disrupts convergence to a consistent 3D reconstruction. Further restricting the setup to only 2 views out of 4 cameras results in unsuccessful training for both latent and explicit models, indicating that very limited camera setups demand careful placement or preliminary encoder pre-training, which should be investigated in a future work.

## B.3 SEGMENTATION MASKS.

| Segmentation | PSNR Future | PSNR Past |
|---|---|---|
| Without masks | 32.66 ± 3.43 | 39.08 ± 3.18 |
| With masks | 33.15 ± 3.51 | 34.55 ± 2.26 |

Table S3: Segmentation Masks

While our final models use segmentation masks for static objects (e.g., ground surfaces), we explore training without these masks to test model reliance on explicit segmentation. We find only a slight reduction in performance (32.66 vs. 33.15), demonstrating the models' capability to implicitly infer static scene regions directly from raw RGB inputs. Thus, explicit segmentation masks are helpful but not strictly essential.

## B.4 MODALITY CONFIGURATIONS

| Modality | PSNR Future | PSNR Past |
|---|---|---|
| 4-1-2-6 | 32.59 ± 3.22 | 33.39 ± 2.21 |
| 4-4-1-3 | 31.98 ± 3.78 | 34.03 ± 2.39 |
| 4-1-1-12 | 33.15 ± 3.51 | 34.55 ± 2.26 |

Table S4: Input Modality

Different input modality configurations were tested. Here, we adapt the notation "a-b-c-d", each varying the temporal span of input conditioning $a$ is the number of particle frames representing the state, $b$ indicates the number of backward frames used to predict $c$ future steps, and total rollout steps during training $d$, leading to $b \cdot c \cdot d$ rollout steps per training step. Both variants ("4-1-2-6", "4-4-1-3" attain competitive performance (PSNR of 32.59 and 32.27 respectively), significantly reducing the computational load compared to longer standard rollouts. This highlights promising avenues for future investigation, emphasizing balance between computational efficiency and performance quality.

## B.5 GRID RESOLUTION

| Grid Size | PSNR Future | PSNR Past |
|---|---|---|
| 0.002 | 25.39 ± 2.95 | 32.28 ± 2.68 |
| **0.004** | **33.15 ± 3.51** | **34.55 ± 2.26** |
| 0.008 | 25.40 ± 3.84 | 30.78 ± 2.63 |
| 0.0012 | 24.06 ± 3.53 | 31.74 ± 2.56 |

Table S5: Grid Resolution.

We test a series of grid resolutions (0.002, 0.004, and 0.008), observing optimal results at 0.004 with a PSNR of 33.15. Both higher (0.008) and finer resolutions (0.002) degrade performance, suggesting an optimal balance achieved at 0.004 between detail preservation and computational complexity for out scene size.

## B.6 Temporal Merger.

| Temporal Merger Setup | PSNR Future | PSNR Past |
|---|---|---|
| [1,1,2,2..] with embedding | 27.55 ± 3.22 | 31.68 ± 2.60 |
| [1,1,4,..] with embedding | 26.79 ± 2.94 | 32.21 ± 2.73 |
| [1,2,2,..] without embedding | 25.07 ± 3.22 | 31.16 ± 2.59 |
| **[1,2,2,..] with embedding (120k)** | **33.15 ± 3.51** | **34.55 ± 2.26** |
| [1,1,1,2,2] with embedding | 18.87 ± 1.50 | 18.09 ± 1.45 |
| [1,4,..] with embedding | 18.19 ± 1.29 | 18.33 ± 1.48 |

Table S6: Temporal Merger. The models are trained for 80k steps if not specified otherwise.

We experiment with various temporal merging strides for each encoder stage combined with embedding options of timestep position encoding. Since we only train with 4 past steps, the strides ".." don't influence the results. After 80k iterations, results clearly indicate two critical factors for success: the use of learned positional embeddings and timing of merging operations. Optimal results occur when merging temporal information only after early spatial processing stages ([1,2,2..]), whereas early or too-late merging drastically diminishes performance. Poor outcomes with late merging likely arise due to spatial pooling operations that dilute vital temporal distinctions before merging.

## C POSITIONING 3DGSIM AMONG EXISTING APPROACHES

To clarify the scope of our contributions, we distinguish 3DGSim from methods focused on dynamic scene reconstruction, and then contrast it with approaches that augment reconstructed 3D scenes with simulation capabilities.

### C.1 CLARIFICATION ON THE DISTINCTION BETWEEN 3DGSIM AND DYNAMIC SCENE RECONSTRUCTION METHODS

Here, we elucidate the fundamental differences between our approach, 3DGSim, and dynamic scene reconstruction methods such as 4DGS Wu et al. (2024a) and DeformableGS Bae et al. (2024).

**Temporal Characteristics:** Dynamic scene reconstruction techniques, including 4DGS and DeformableGS, aim to reconstruct a temporally-varying 3D representation from a complete set of video frames, encompassing past, present, and future data. These methods are inherently temporally *non-causal*, as they leverage the entire video sequence to infer and optimize a 4D (3D + time) representation of the observed events. Their primary function is to interpolate or reconstruct what has already occurred within the video.

**Predictive vs. Reconstructive Nature:** In stark contrast, 3DGSim is designed for temporally *causal prediction*. Given a set of scene representations from a few past timesteps, our model's objective is to predict the future evolution of the scene. This prediction is made without any access to future video frames. This predictive capability necessitates that the model develops an understanding of the underlying physics governing the scene's dynamics.

### C.2 COMPARISON TO PHYSGAUSSIAN AND PHYSDREAMER

PhysGaussian Xie et al. (2023) and PhysDreamer Zhang et al. (2024b) reconstruct 3D representations from multi-view images of static scenes and then simulate mesh-node dynamics via the Material Point Method (MPM). While these methods are effective for VR and per-scene content creation, a direct comparison to 3DGSim is not feasible for several reasons:

**PhysGaussian does not learn dynamics.** PhysGaussian reconstructs a 3D representation from a *static* scene and simulates particles with MPM. In our experiments, training images are generated using a combination of MPM simulators. One could attempt to tune the PhysGaussian MPM simulator to match the data-generating simulation, but hand-tuning MPM parameters to match the motion of another simulation is notoriously difficult. In contrast, 3DGSim learns to simulate object dynamics

directly from video without requiring physical priors. *While our method can be extended to learn different material modalities and collisions, it is unclear how to tackle this problem with MPM.*

**PhysDreamer performs per-scene optimization.** Our method is a feed-forward world model trained once and reused across scenes. By contrast, PhysDreamer requires training a separate model for each trajectory.

**PhysDreamer does not simulate collisions.** As the authors note: "In this work, we restrict our scope to elastic objects without collisions." In our experiments, 3DGSim accurately simulates both elastic and rigid collisions.

**In PhysDreamer (and MPM in general), boundary conditions must be defined by hand.** Static parts of objects are manually set to be static. 3DGSim learns constrained dynamics directly from videos.

**MPM requires very small simulation steps, making full-trajectory backpropagation impractical.** For instance, in PhysDreamer the timestep is $\Delta t = 1 \times 10^{-4}$ with 768 sub-steps per frame. Training is split into two stages: 1) optimizing initial velocities on a few frames, 2) estimating material parameters with fixed velocities. In the second stage, gradients are truncated to flow only one frame backward to avoid explosion/vanishing gradients. As the authors state Zhang et al. (2024b):

> "Rather than optimizing the material parameters and initial velocity jointly, we split the optimization into two stages for better stability and faster convergence. In particular, in the first stage, we randomly initialize the Young's modulus for each Gaussian particle and freeze it. We optimize the initial velocity of each particle using only the first three frames of the reference video. In the second stage, we freeze the initial velocity and optimize the spatially varying Young's modulus. During the second stage, the gradient signal only flows to the previous frame to prevent gradient explosion/vanishing."

In contrast, 3DGSim jointly learns 3D reconstruction and dynamics simulation in a *fully end-to-end* manner, with *backpropagation through entire trajectories*, without requiring staged training or gradient truncation. Its transformer-based architecture enables significantly larger timesteps. Empirically, 3DGSim achieves accurate long-range predictions while being *1–2 orders of magnitude faster to train and simulate*.

### C.3 SUMMARY:

The objective of methods like 4DGS and DeformableGS is scene reconstruction. They are not designed to forecast how a physical scene will evolve. Conversely, 3DGSim is a particle-based simulator that learns physics solely from video, enabling forecasting. Approaches like PhysGaussian and PhysDreamer reconstruct 3D representations from static scenes and then use material-point method (MPM) for simulation, they do not incorporate learning of the dynamics in the same manner. In fact, the authors of PhysGaussian emphasize that their framework is unable to handle collisions, a key aspect of physical simulation.

Our work instead tackles vision-based physics learning, which poses a distinct challenge beyond dynamic 3D scene reconstruction. Because of this difference, no suitable baselines currently exist, and we therefore benchmark against video generation models.

# D VISUALIZATIONS

In this section we show rolled out dynamic prediction for different scenarios. If not otherwise stated, the first row in each image shows the ground truth predictions. First we depict scene editability and composabilty followed by several generalizations to multi body and visualization for each dataset. Finally we depict several rollouts from the test set for every type of dynamcis for both 3DGSim and CosmosFT. CosmosFT is conditioned on past 5 frames and following prompts:

| Prompt |
| --- |
| *A rigid body falling on a circular gray ground* |
| *A soft body falling on a circular-gray-ground* |
| *A rigid body falling on a red rectangular cloth which is fixed on its corners* |

Table S7: Prompts used to condition Cosmos and CosmosFT.

**Videos of the predictions below are available in the supplementary material.**

## D.1 SCENE EDITABILITY

### D.1.1 3DGSIM

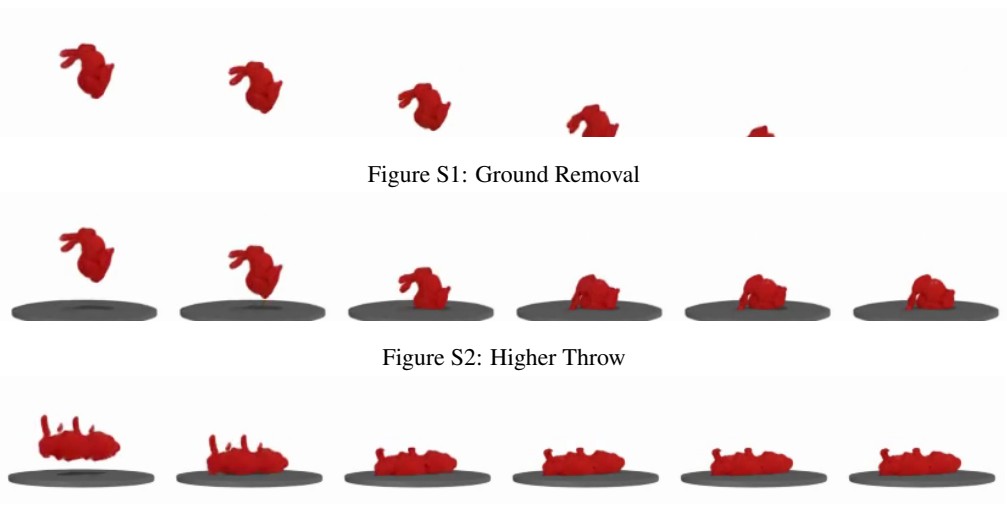

Figure S1: Ground Removal

Figure S2: Higher Throw

Figure S3: Parallel Simulation

### D.1.2 COSMOSFT

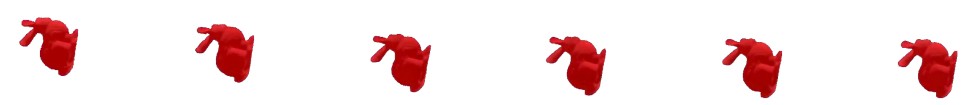

Figure S4: CosmosFT: When ground is removed, objects often remain suspended.

## D.2 GENERALIZATION TO MULTIPLE BODIES

### D.2.1 3DGSIM

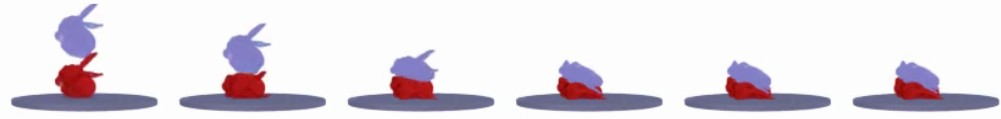

Figure S5: Scene with 2 bunnies.

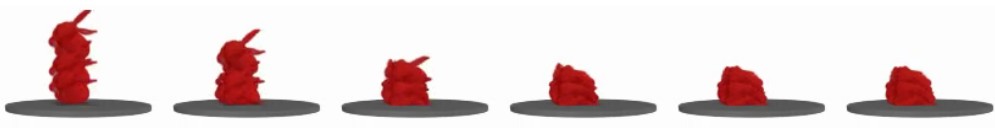

Figure S6: Scene with 3 bunnies.

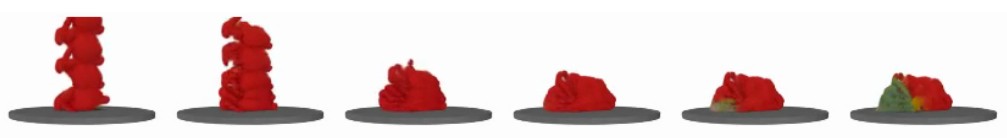

Figure S7: Scene with 5 bunnies. Interestingly, the combined weight of five objects slowly pushes the particles through the table causing color changes.

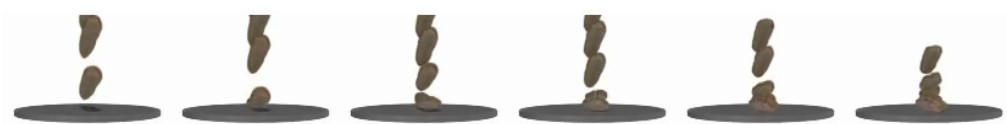

Figure S8: Scene with 5 worms.

### D.2.2 COSMOSFT

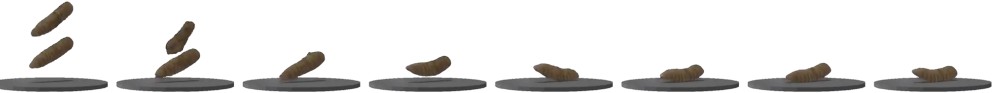

Figure S9: Cosmos FT: multiple worms are morphed into one single object before colliding with the ground.

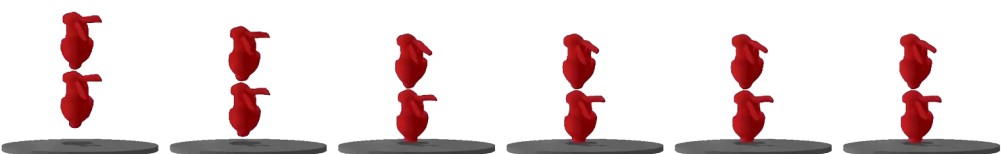

Figure S10: CosmosFT: two bunnies levitate above each just before colliding with the ground.

## D.3 VISUALIZATION OF CLOTH SIMULATIONS

Here we present various cloth simulations for different objects. These are all trajectories from the test-set. The first row is the ground-truth and the second row the prediction.

### D.3.1 3DGSIM

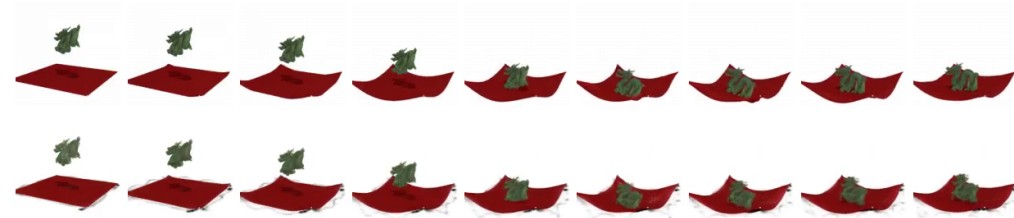

Figure S11: Dragon

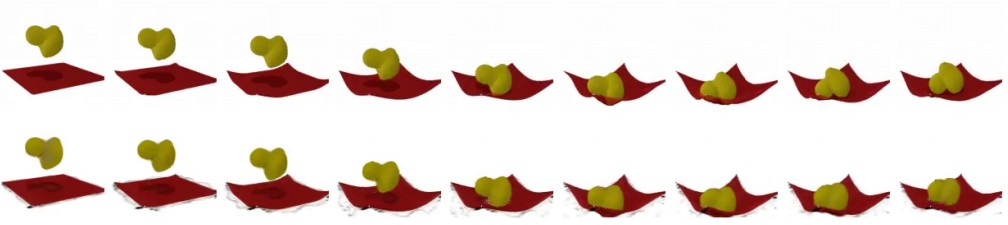

Figure S12: Duck

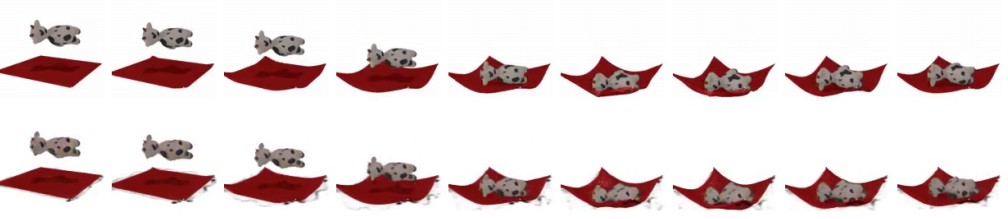

Figure S13: Cow

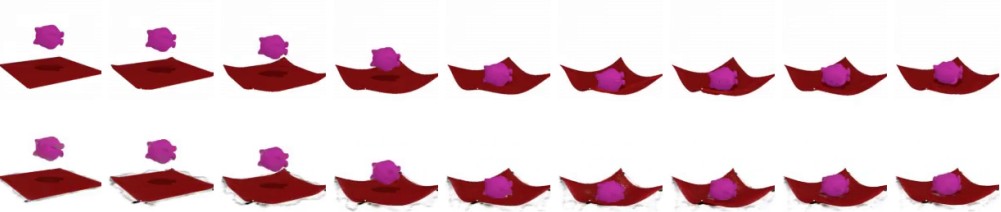

Figure S14: Devil

### D.3.2 COSMOSFT

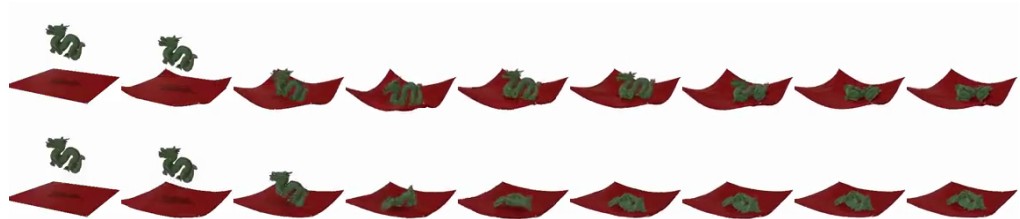

Figure S15: Dragon

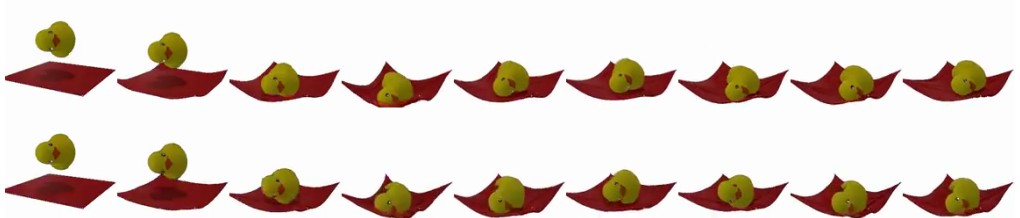

Figure S16: Duck

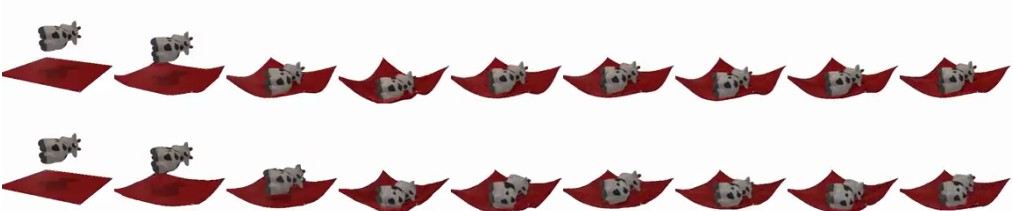

Figure S17: Cow

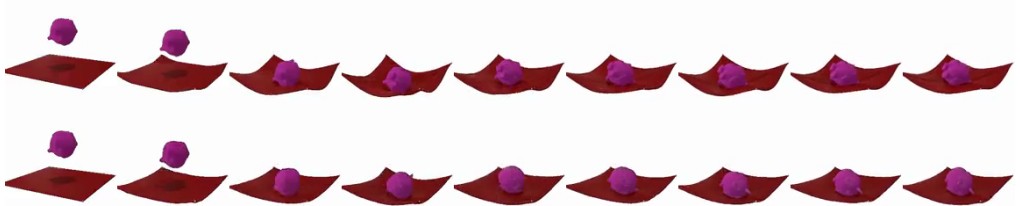

Figure S18: Devil

## D.4 VISUALIZATION FOR ELASTIC DYNAMICS

Here we present various elastic simulations for different objects. These are all trajectories from the test-set. The first row is the ground-truth and the second row the prediction.

### D.4.1 3DGSIM

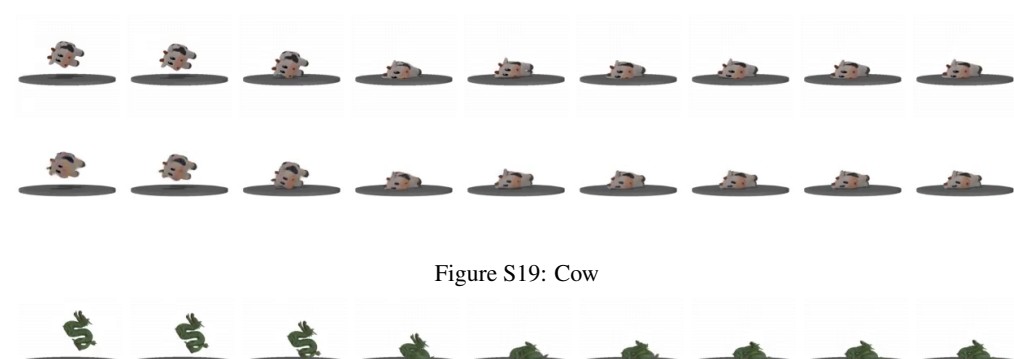

Figure S19: Cow

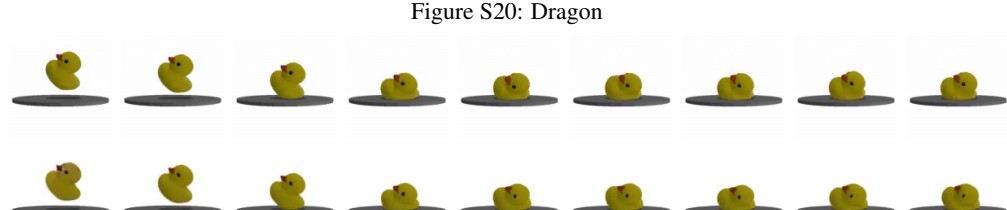

Figure S20: Dragon

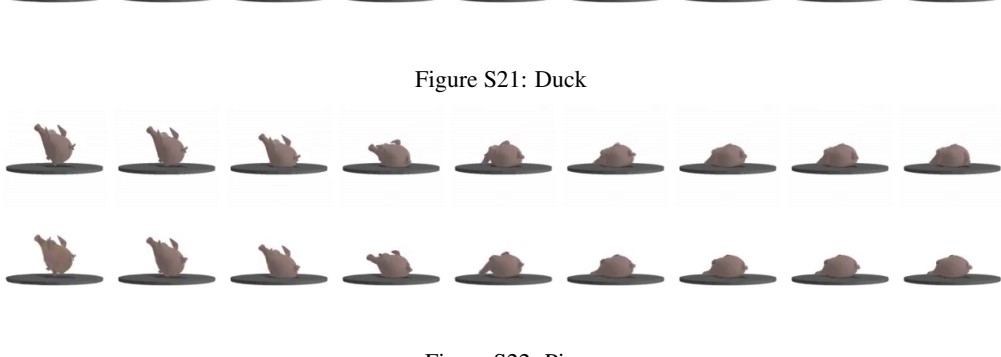

Figure S21: Duck

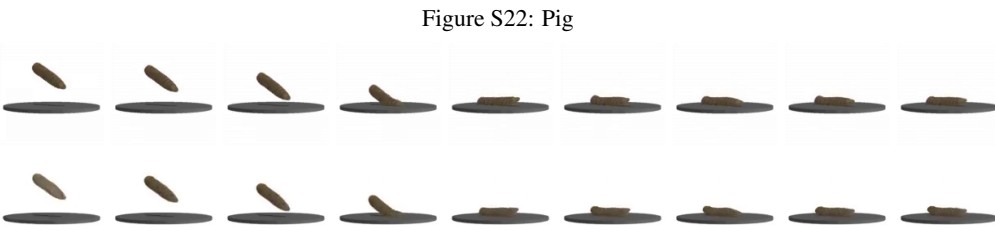

Figure S22: Pig

Figure S23: Worm

### D.4.2 COSMOSFT

Figure S24: Cow

Figure S25: Dragon

Figure S26: Duck

Figure S27: Pig

Figure S28: Worm

## D.5 VISUALIZATION FOR RIGID BODY DYNAMICS

Here we present various rigid simulations for different objects. These are all trajectories from the test-set. The first row is the ground-truth and the second row the prediction.

### D.5.1 3DGSIM

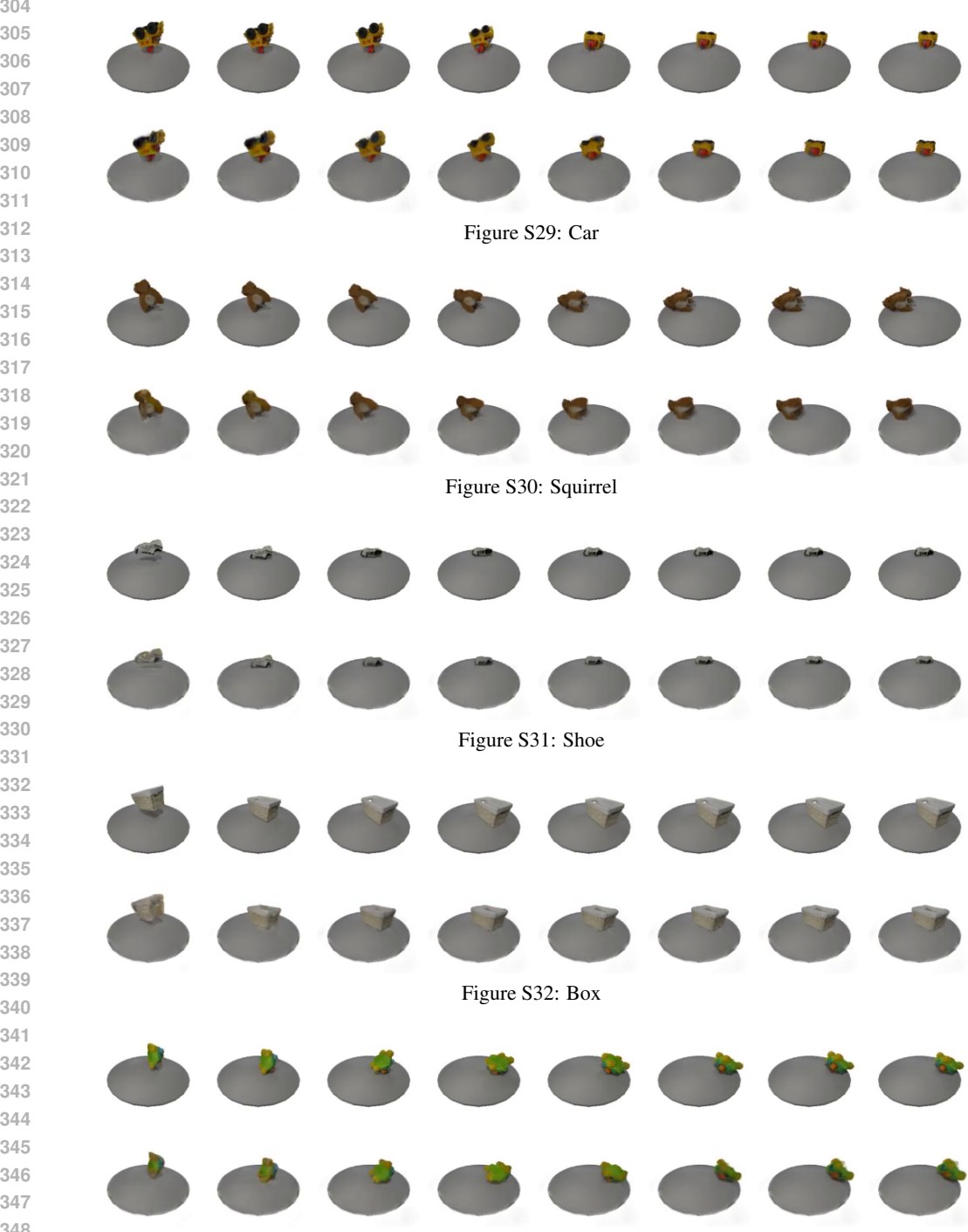

Figure S29: Car

Figure S30: Squirrel

Figure S31: Shoe

Figure S32: Box

Figure S33: Turtle

### D.5.2 COSMOSFT

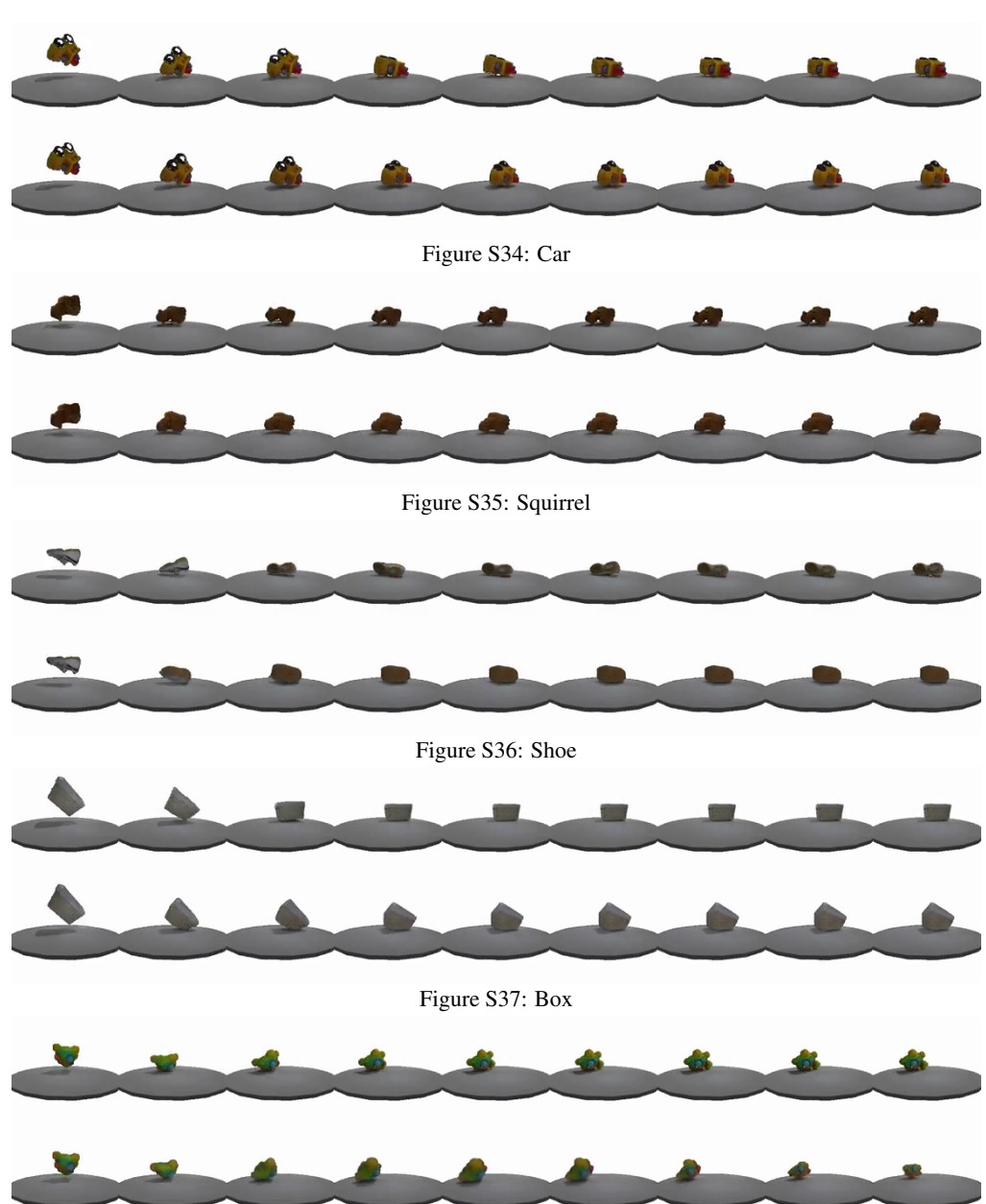

Figure S34: Car

Figure S35: Squirrel

Figure S36: Shoe

Figure S37: Box

Figure S38: Turtle

