# OpenReview forum: "Learning 3D-Gaussian Simulators from RGB Videos"
_ICLR.cc/2026/Conference — Submitted to ICLR 2026_

### Official Review · Reviewer_jav5 · 2025-10-28

**Soundness:** 3
**Presentation:** 2
**Contribution:** 2
**Rating:** 4
**Confidence:** 3

**Summary:**

This paper introduces 3DGSim, an end-to-end differentiable framework that learns 3D physical simulation directly from multi-view RGB videos. 3DGSim combines a feed-forward inverse renderer based on MVSplat for reconstructing 3D Gaussian particles, a transformer-only dynamics model leveraging space-filling curves and temporal merging for efficient spatiotemporal reasoning, and a Gaussian splatting renderer for differentiable image supervision. Trained solely with image reconstruction loss, the model learns latent visuo-physical particle features that capture diverse dynamics without explicit physics priors. Experiments demonstrate that 3DGSim achieves high-fidelity, physically plausible long-horizon predictions, robust generalization to unseen scenarios such as ground removal or multi-object interactions.

**Strengths:**

-  The model generalizes to unseen scenarios such as ground removal and multi-object interactions, demonstrating robust physical understanding and scene editability.
-  Extensive experiments across rigid, elastic, and cloth datasets show physically plausible long-horizon rollouts with strong quantitative and qualitative performance over 2D baselines like Cosmos.

**Weaknesses:**

- The paper does not provide quantitative comparisons with existing 3D physical simulators. Although the authors mention that code and data for these baselines are unavailable, the absence of such comparisons weakens the rigor and positioning of the work.
- Several implementation and presentation details are omitted or unclear, which may cause confusion for readers:
  - The variable $p$ in Equation (2) is not explicitly defined.
  - The process of extracting the feature $f$ is not clearly explained.
  - The paper does not specify what dataset(s) were used for training or how many data samples are included.
  - The meaning of “Groundtruth” in Figure 7 is unclear.
  - It is not explained how the ground plane is represented or modeled in the method.
- All experiments are on synthetic datasets. The model's performance on real-world videos remains to be verified.
- It would be better if the author could add some ablation study of key design parts.
- Missing related work: The proposed idea is also related to [1].

[1] Latent Intuitive Physics: Learning to Transfer Hidden Physics from A 3D Video. ICLR 2024.

**Questions:**

- Part of the questions are listed in the weakness part.
- In Figure 12, what exactly do “input” and “reconstruction” views refer to? Is the “12 (4+5)” notation a typo or intentional (since 4+5 ≠ 12)?
- In Figure 14, both “inference speed” and “cloth/rigid data FPS” are reported. What is the difference between inference FPS and dataset FPS? Is there any difference in inference FPS between the two dataset?

---

> ### Author Response · Authors · 2025-11-20
>
> Thank you for the helpful feedback and for the positive assessment of our method. We are glad that the experiments clarified the impact of our design choices, and we will improve the clarity of these results in the revision. We also appreciate the reviewer’s recognition of our counterfactual experiments, which highlight how the 3D formulation leads to stronger generalization than the 2D alternative.
>
> ---
>
> > It would be better if the author could add some ablation study of key design parts.
>
> We agree that clearer ablations would strengthen the paper. Although some results are already spread across the paper and appendix, we will add a concise bar plot where each variant removes one key component (e.g., TEM and attention strategy, feature representation, rollout length, modality setup,..). Would this addition be sufficient to show the contribution of each part?
>
> > The variable p in Equation (2) is not explicitly defined.
>
> Thank you for catching these potential sources of confusion in our paper. p is the xyz-position (as introduced in equation 1), i and t_k denote the particle index and timestep. We will improve the clarity of the used notation in the revised paper.
>
> > The process of extracting the feature is not clearly explained.
>
> In Subsection 4.2 we explain how MVSplat is extended to extract the feature, whereas MVSplat is introduced in Section 3.  Could you please provide more details on what we could change to improve the explanation of the feature extraction?
>
> > The paper does not specify what dataset(s) were used for training or how many data samples are included.
>
> In Chapter 5 we mention both: datasets have 200 trajectories per object and 12% of them are held out for testing. We will provide further details on how the datasets were generated in the Appendix of the revised work.
>
> > The meaning of “Groundtruth” in Figure 7 is unclear.
>
> The first and last column in figure 7 represent the initial and last frame of the “true” simulated motion. The in-between columns/frames are predictions of 3DGSim. The last two frames should look similar. We will further clarify this in the revised paper.
>
> > It is not explained how the ground plane is represented or modeled in the method.
>
> No special treatment is done for the ground plane. It is treated the same as the rest of the scene, i.e. it also consist of a set of particles which the model has learned to keep fixed.
>
> > Missing related work: The proposed idea is also related to [1] Latent Intuitive Physics.
>
> We thank the reviewer for bringing the paper in [1] to our attention. Even though it only concerns fluids, it is related and we will add it to the related work in the revision.
>
> > In Figure 12, what exactly do “input” and “reconstruction” views refer to? Is the “12 (4+5)” notation a typo or intentional (since 4+5 ≠ 12)?
>
> From the 12 views we first randomly sample 4 views as context / input (without replacement), then 5 views as target during training
>
> > In Figure 14, both “inference speed” and “cloth/rigid data FPS” are reported. What is the difference between inference FPS and dataset FPS? Is there any difference in inference FPS between the two dataset?
>
> Dataset FPS is the framerate of the ground-truth simulations (42 FPS for cloth/elastoplastic, 12 FPS for rigid). Inference FPS is how fast 3DGSim runs on a GPU. The colored bars show inference speed, while the interrupted line marks the simulation FPS required for real-time. There is no difference in inference FPS between datasets, the variation comes only from using 4 vs. 2 past steps
>
> Just as a side note, the underlying simulators for creating the dataset run at much higher internal rates, but we downsample to the /dataset FPS/.

---

### Official Review · Reviewer_qMN5 · 2025-10-30

**Soundness:** 2
**Presentation:** 3
**Contribution:** 3
**Rating:** 4
**Confidence:** 3

**Summary:**

This paper introduces 3DGSim, a fully differentiable framework that learns 3D particle-based simulators from multi-view RGB videos. The method integrates inverse rendering, transformer-based dynamics, and Gaussian splatting for novel-view synthesis. The authors demonstrate the model's ability to simulate rigid, elastic, and cloth-like dynamics and show generalization to out-of-distribution scenarios such as ground removal and multi-object interactions. However, the experimental evaluation lacks comprehensive comparisons with relevant 3D baselines, and the presented results appear limited in complexity and realism.

**Strengths:**

1. The proposed framework is fully differentiable and learns directly from multi-view RGB videos, eliminating the need for privileged information such as depth or object tracks.
2. The method introduces a transformer-only dynamics model that avoids hand-crafted graph structures and uses space-filling curves for efficient spatiotemporal reasoning.
3. The paper is overall well-written and should be easy to follow.

**Weaknesses:**

1. There is a lack of quantitative and qualitative comparisons with relevant 3D baselines (e.g., PhysGaussian, or other particle-based simulators). The evaluation is primarily limited to a comparison with Cosmos, a 2D video generation model. While this highlights differences in representation (2D vs. 3D), it does not adequately demonstrate the advantages of the proposed method over existing 3D approaches that tackle similar tasks.
2. The presented examples appear simplistic and lack realism. It is unclear how well the method generalizes to real-world data. Evaluation on established real-world benchmarks (e.g., PhysGaussian’s dataset) would strengthen the claims.
3. The experiments are limited primarily to falling objects. It is important to demonstrate the method’s performance on a wider variety of dynamic scenes (e.g., complex collisions, articulated objects or robot manipulation).

**Questions:**

The paper template appears inconsistent: the title should be left-aligned, and the header should include "Under review as a conference paper at ICLR 2026".

---

> ### Author Response · Authors · 2025-11-20
>
> Thank you for your time reviewing our work helping us to improve the overall quality of the manuscript. We are grateful that the removal of privileged information requirements and the design of our transformer-only dynamics model were well received, and that the paper is found clear and easy to follow.
>
> ---
>
> > There is a lack of quantitative and qualitative comparisons with relevant 3D baselines (e.g., PhysGaussian, or other particle-based simulators). The evaluation is primarily limited to a comparison with Cosmos, a 2D video generation model. While this highlights differences in representation (2D vs. 3D), it does not adequately demonstrate the advantages of the proposed method over existing 3D approaches that tackle similar tasks.
>
> We thank the reviewer for emphasizing the importance of situating 3DGSim among 3D approaches. PhysGaussian (and the series of works from there) are indeed impressive works, but their goals differ substantially from ours. They reconstruct static 3D scenes/object from multi-view images, separate the object, manually specify ground boundary conditions, and then simulate motion using hand-tuned MPM parameters. In contrast, 3DGSim learns dynamics directly from videos in a fully feed-forward, end-to-end manner, requiring no physical priors, no manual boundary conditions, and no per-scene optimization. Moreover, PhysDreamer (feed-forward parameter estimation method based on PhysGaussian) explicitly limits itself to elastic objects without collisions, while 3DGSim handles both elastic and rigid interactions.  Given these fundamental differences, we are unsure how these methods should be qualitatively compared.
>
> We may be misunderstanding the reviewer’s suggestion, and there may be a comparison angle we have overlooked. We would greatly appreciate clarification on what type of evaluation or baseline the reviewer believes would yield meaningful insights.
>
> *[0] PhysGaussian: Physics-Integrated 3D Gaussians for Generative Dynamics, CVPR 2024* \
> *[1] PhysDreamer: Physics-Based Interaction with 3D Objects via Video Generation, ECCV 2024*
>
> > Evaluation on established real-world benchmarks (e.g., PhysGaussian’s dataset) would strengthen the claims.
>
> Thank you for this suggestion. After looking more closely into the datasets used in PhysGaussian (e.g., BlenderNeRF, Instant-NGP, Nerfstudio, and the DroneDeploy NeRF scenes), we found that these are all static reconstruction datasets that PhysGaussian post-simulates, as explained earlier. Because they do not contain actual dynamic interactions, they are not directly suitable for evaluating real-world generalization in our setting.
>
> If the reviewer is aware of any real-world datasets that contain ground-truth physical dynamics without actions suitable for benchmarking simulation methods, we would like to add them to the experiments of the revised work.
>
> > The experiments are limited primarily to falling objects. It is important to demonstrate the method’s performance on a wider variety of dynamic scenes (e.g., complex collisions, articulated objects or robot manipulation).
>
> We agree that our current experiments focus on falling-object scenarios, but the framework is designed to generalize beyond them. By making it publicly available, we invite the community and ourselves to explore richer dynamic settings, including articulated motion and further action conditioning, which are essential for manipulation.
>
> > The paper template appears inconsistent: the title should be left-aligned, and the header should include "Under review as a conference paper at ICLR 2026".
>
>
>
> Thank you for helping us to improve the paper’s formatting. We will correct the title and header in the revised version

---

### Official Review · Reviewer_X9bZ · 2025-10-30

**Soundness:** 3
**Presentation:** 3
**Contribution:** 2
**Rating:** 4
**Confidence:** 3

**Summary:**

The paper proposes 3DGSim, an end-to-end differentiable pipeline that learns a particle-based 3D simulator directly from multi-view RGB videos. Experiments on rigid, elastoplastic, and cloth regimes compare against strong 2D video predictors and include ablations over state parameterization, camera count, and masking, alongside a speed study indicating near real-time rollouts on an H100.

**Strengths:**

1. The implementation details are extensive, promoting its reproducibility.
2. The three-module architecture is laid out crisply, the motivation for replacing kNN with serialization is well argued, and figures for temporal merging/serialization make the mechanism concrete.
3. The paper is well written and easy to follow.

**Weaknesses:**

1. The comparison of some related open-sourced works is missing:

  [1] NeuMA: Neural Material Adaptor for Visual Grounding of Intrinsic Dynamics

  [2] PAC-NeRF: Physics Augmented Continuum Neural Radiance Fields for Geometry-Agnostic System Identification

  [3]  Vid2Sim: Generalizable, Video-based Reconstruction of Appearance, Geometry and Physics for Mesh-free Simulation

2. Physical fidelity is assessed primarily with image metrics (PSNR/SSIM/LPIPS) derived from reconstruction loss, so it remains unclear how well the model captures contact timing, energy dissipation, or material parameters.
3. The datasets are fully simulated, and although the authors plan real-world validation, current claims about real-scene deployment are necessarily tentative.

**Questions:**

1. To help the paper reach its stated goals, I would encourage the authors to add at least a small set of physics-grounded metrics—contact event timing, restitution estimates, mass/Young’s modulus recovery, or energy drift—alongside PSNR/LPIPS/SSIM, which would substantiate the “learned physics” claim beyond photometric fidelity.
2. On modeling and analysis, I’m curious whether the latent features correlate with interpretable physical parameters. A simple probing experiment (or a lightweight regressor trained post-hoc) could test identifiability for density or stiffness using the simulated ground truth.

---

> ### Author Response · Authors · 2025-11-20
>
> We are glad the reviewer highlighted the benefits of removing privileged information, the strength of our transformer-only dynamics design, and the overall clarity of the paper.
>
> ---
>
> > To help the paper reach its stated goals, I would encourage the authors to add at least a small set of physics-grounded metrics—contact event timing, restitution estimates, mass/Young’s modulus recovery, or energy drift—alongside PSNR/LPIPS/SSIM, which would substantiate the “learned physics” claim beyond photometric fidelity.
>
> We thank the reviewer for the suggestion to help improve the paper.
>
> Regarding physics-grounded metrics: we see two main directions in learned simulation research: (i) **parameter-recovery approaches** and (ii) **fully data-driven approaches**. The first category includes works such as PhysDreamer, PAC‑NeRF, NeuMA etc, which perform per-scene optimization to recover pre-defined simulation parameters under known boundary conditions and material assumptions (e.g., elasticity, rigidity). In contrast, our method belongs to the second category, where the dynamics are learned directly from videos in an implicit, end‑to‑end fashion without explicit parameter recovery. Examples of this direction include video-generation methods such as *Cosmos [4]* and *Learning 3D Particle-based Simulators from RGB‑D Videos [5]*.  Consequently, quantities like mass or Young’s modulus are not directly accessible within our learned latent space. We appreciate the reviewer’s point, and would be glad to explore whether approximate proxies, such as energy consistency or contact timing, could be estimated post‑hoc. If the reviewer has a specific suggestion for how to define such metrics in a data‑driven setting, we would be very grateful for guidance.
>
> >The comparison of some related open-sourced works is missing: \
> [1] NeuMA: Neural Material Adaptor for Visual Grounding of Intrinsic Dynamics \
> [2] PAC-NeRF: Physics Augmented Continuum Neural Radiance Fields for Geometry-Agnostic System Identification \
> [3] Vid2Sim: Generalizable, Video-based Reconstruction of Appearance, Geometry and Physics for Mesh-free Simulation
>
> Concerning related work NeuMA [1], PAC‑NeRF [2], Vid2Sim [3], SpringGauss [6], and PhysDreamer [7] all rely on per‑scene optimization with manually engineered components such as differentiable backend simulators, material‑specific parameters (e.g., Young’s modulus and Poisson’s ratio for elasticity or viscosity for fluids), and predefined boundary conditions (ground, fixed points). These methods reconstruct the 3D object (the rest of the scene is set as boundary condition) from multi‑view static images and then simulate mesh‑node dynamics via MPM. By contrast, 3DGSim is a feed‑forward model trained once and reused across scenes, learning to “simulate what it sees” without explicit physical modeling of different materials or boundary conditions.
>
> We can include a qualitative comparison table summarizing these distinctions to make this clearer in the revision.  We would also welcome further feedback. Could the reviewer advise on how to design a meaningful quantitative comparison across such different paradigms?
>
> *[4] Cosmos world foundation model platform for physical AI, 2025* \
> *[5] Learning 3D Particle-based Simulators from RGB‑D Videos, ICLR 2024* \
> *[6] Reconstruction and Simulation of Elastic Objects with Spring-Mass 3D Gaussians* \
> *[7] PhysDreamer: Physics-Based Interaction with 3D Objects via Video Generation, ECCV 2024*
>
> > On modeling and analysis, I’m curious whether the latent features correlate with interpretable physical parameters. A simple probing experiment (or a lightweight regressor trained post-hoc) could test identifiability for density or stiffness using the simulated ground truth.
>
> We thank the reviewer for this interesting suggestion regarding representation analysis. Since our dataset includes diverse material types (rigid, elastoplastic, and cloth), there is no unified set of physical parameters across them. In fact different MPM-based simulators are used to generate the data. These simulators employ distinct parameterizations for each material category, which makes linear probing for such quantities non-trivial. As an alternative, we could explore unsupervised analyses, e.g., PCA or t-SNE of the learned 3D particle features, to examine whether clusters emerge that correspond to material types or physical behaviors. Would such an analysis align with what the reviewer had in mind?

---

> > ### Comment · Reviewer_X9bZ · 2025-11-25
> >
> > Thanks to the authors for the detailed and thoughtful rebuttal.
> >
> > > Physical metric
> >
> > I understand that analyzing such physical parameters is quite difficult for the current pipeline. However it is feasible to conduct an experiment comparing the contact event timing of particle groups between the simulated results and the ground truth. (Though the method learns only from videos, the videos are still recorded from a simulator to my knowledge)
> >
> > > Related work comparison
> >
> > Vid2Sim has two stages. The first stage is generalizable to different scenes. Could you provide some comments on comparing 3DGSim with Vid2Sim's first stage?
> >
> > > Linear Probe
> >
> > I now understand this point and have no further questions regarding this topic.

---

> > > ### Author Response · Authors · 2025-11-26
> > >
> > > Thank you for the thoughtful follow-up and for highlighting these points.
> > >
> > > >I understand that analyzing such physical parameters is quite difficult for the current pipeline. However it is feasible to conduct an experiment comparing the contact event timing of particle groups between the simulated results and the ground truth. (Though the method learns only from videos, the videos are still recorded from a simulator to my knowledge)
> > >
> > > For the current dataset, we only stored videos, but we can generate additional trajectories (with privileged simulator state access) to support this experiment.
> > >
> > > Our understanding is that the reviewer refers to the timing of the first contact between the object and the ground particle groups, i.e., the first frame in which the distance between any particles from the two groups falls below a small threshold.
> > >
> > > Since 3DGSim does not explicitly model contact or contact forces but instead learns them implicitly through spatial attention, we need a robust definition of contact that avoids sensitivity to outlier or semi-transparent particles.
> > >
> > > Our current plan is to:
> > > - split particles into two groups (object and ground),
> > > - define contact as the centers of two Gaussians being within a distance threshold,
> > > - enforce robustness by declaring contact only when a small fraction of particles satisfy this condition (e.g., 0.5%–2%),
> > > - record the first frame in which this occurs.
> > >
> > > Does this align with what the reviewer is proposing? \
> > > We are also happy to adopt a standard metric if there is an established paper or methodology you would recommend.
> > >
> > >
> > > > Vid2Sim has two stages. The first stage is generalizable to different scenes. Could you provide some comments on comparing 3DGSim with Vid2Sim's first stage?
> > >
> > > **Reconstruction** \
> > > Vid2Sim reconstructs only the segmented object, while the remainder of the scene is excluded from reconstruction; for instance, the floor is not recovered from the video but is manually added in the simulator as a boundary condition.
> > >
> > > 3DGSim reconstructs the whole scene as a unified representation and also does not rely on manual boundary-setting.
> > >
> > > **Learning** \
> > > Vid2Sim’s physical parameter retrieval assumes a known constitutive model (e.g., Neo-Hookean with Young’s modulus, Poisson’s ratio, and LBS weights) and is trained using ground-truth physical parameter labels. In addition, it predicts a single scalar parameter per object that is shared across all points.
> > >
> > > In contrast, 3DGSim does not rely on ground-truth physics labels, does not assume a specific constitutive model, and can represent spatially varying properties, since each particle carries its own latent state.
> > >
> > > **Simulation** \
> > > Vid2Sim uses the same simulator for generating and fitting the data (i.e., simulator-in-the-loop).
> > >
> > > 3DGSim is trained independently of the simulator used to produce the data (in our case 3 different ones).
> > >
> > > **Overall framing / design tradeoffs** \
> > > Both methods have different tradeoffs. Vid2Sim adopts strong inductive bias for the observed dynamics to deliver efficient and reliable per-scene simulator recovery under those assumptions.
> > >
> > > Whereas 3DGSim gives up efficient (per-scene) learning and, inspired by [1], proposes a model that fits the idea of large scale pretraining and generalization to different types of physics.
> > >
> > > [1] Learning to Simulate Complex Physics with Graph Networks

---

> > > > ### Author Response · Authors · 2025-12-01
> > > > **Time of Impac**
> > > >
> > > > ## Contact timing (X9bZ)
> > > >
> > > > In our previous response, the reviewer proposed measuring the **Time of Impact** to quantify the physical plausibility of our model beyond visual metrics. According to the proposed threshold contact detection we have now computed the absolute frame error between the predicted contact time and the ground truth. For this we generated 10  trajectories per object with privileged 3D simulator state.
> > > >
> > > > **Results (Absolute Frame Error):**
> > > >
> > > > - **Elastic:** $0.69 \pm 0.72$
> > > > - **Cloth:** $1.10 \pm 1.13$
> > > > - **Rigid:** $1.36 \pm 1.11$
> > > >
> > > > **Discussion of Results**
> > > >
> > > > The results indicate that 3DGSim predicts impact timing with an average error of approximately **1 frame**.
> > > >
> > > > However, we found this metric to be highly sensitive to reconstruction quality. Because 3DGSim learns an **implicit** notion of physics rather than using a hard-coded collision engine, small artifacts (e.g., semi-transparent floaters near the ground) can falsely trigger the distance threshold. While the low error still indicates physically grounded dynamics, we view visual fidelity and temporal consistency as more reliable metrics for end-to-end learned simulators without manually defined boundaries.
> > > >
> > > > *Note:* We added a KNN-based filtering step that removes such artifacts by analyzing local particle cluster statistics.

---

### Official Review · Reviewer_KeGc · 2025-10-31

**Soundness:** 3
**Presentation:** 2
**Contribution:** 3
**Rating:** 6
**Confidence:** 4

**Summary:**

This work proposes a 3DGS-based physical simulation framework that accurately models really visual-physical processes. It integrates an inverse renderer built on MVSplat, a Transformer-only dynamics module without explicit neighborhood graphs, and a differentiable 3D Gaussian Splatting renderer. Trained with a joint reconstruction-and-prediction objective, the system learns geometry, appearance, and dynamics simultaneously. The authors validate long-horizon predictions and novel-view consistency under OOD settings, such as terrain changes, multi-body interactions, and lighting across rigid, elastoplastic, and cloth scenarios, and show stronger visual metrics and near real-time inference compared to 2D video baselines. Overall, the framework unifies physically consistent 3D generation with interpretable particle states in a single framework, addressing inherent limitations of 2D generation and laying a potential foundation for downstream applications.

**Strengths:**

1.	The framework maintains stable, physically consistent rollouts under diverse conditions and generalizes well: even when trained only on "single-object-ground contact," it produces plausible multi-body interactions and reproduces visual phenomena such as shadows.
2.	By leveraging PTv3, it avoids the overhead of kNN graph construction and distance computation, enabling near real-time inference.
3.	It is validated on multiple datasets and can effectively learn material dynamics even from limited samples.

**Weaknesses:**

I have some concerns about 3DGSim:
1.	The experiments lack ablations on the TEM mechanism. If TEM is replaced with conventional temporal stacking/causal attention, how much does 3DGSim's performance degrade?
2.	Without explicit physical constraints in optimizing the dynamics model, how is to prevent interpenetration between particles?
3.	Although 3DGSim performs strongly on synthetic datasets, there is no validation on real data. Can it retain its performance under lighting changes and real-world scenarios? In addition, what are the computational and data requirements (for both training and inference) that constitute the practical threshold for deployment?

**Questions:**

see weakness

---

> ### Author Response · Authors · 2025-11-20
>
> Thank you for your time. We are grateful for the helpful remarks and the positive assessment of our work.
>
> ---
>
> > The experiments lack ablations on the TEM mechanism. If TEM is replaced with conventional temporal stacking/causal attention, how much does 3DGSim's performance degrade?
>
> Thank you for pointing out how to further improve the ablations. . We actually included such an ablation in the last row of Table B.6, where we replace TEM with conventional attention across all timesteps. This leads to a substantial performance drop (PSNR 33.15 → 18.19), indicating that TEM plays a crucial role. We note that this baseline still uses the spatially hierarchical attention structure of PTv3. Running full attention over all spatial points would indeed be an informative comparison, but is computationally prohibitive given the scale of point clouds (quadruples memory peaks). We will revise the ablation on the TEM layer to clarify which setting corresponds to conventional causal attention.
>
> > Without explicit physical constraints in optimizing the dynamics model, how is to prevent interpenetration between particles?
>
> There are no explicit guarantees that interpenetrations do not occur. However, our visual results, including counterfactual tests (removed ground, repositioned, and interacting objects), suggest that the model has learned a contact‑aware representation that produces realistic, stable interactions. We attribute this to the “spatial causality” inherent in a fully 3D representation, which encodes spatial relationships between particles via spatially-local self-attention, unlike 2D models such as COSMOS (or 2D+1 RGB‑D approaches).
>
> > Although 3DGSim performs strongly on synthetic datasets, there is no validation on real data. Can it retain its performance under lighting changes and real-world scenarios?
>
> To definitely answer these questions, a large scale training should be performed, which we could not yet do. This is due to lack of open multiview datasets and computational constraints. However, our architecture should generalize to lightning conditions because we use latent features in our pipeline, which have been demonstrated to become invariant to many nuances if trained at scale. For instance, Depth Anything 3 [1] shows strong invariance to lighting variation, including on the unseen HiRoom [2] benchmark with spatially varying HDRI illumination. Instead of training all end to end, we can use such a pretrained feature extractor in our pipeline, which would be promising direction for future work.
>
> *[1] Depth Anything 3: Recovering the Visual Space from Any Views* \
> *[2] Svlightverse: Large-scale photorealistic indoor dataset with spatially-varying hdri lighting*
>
> > In addition, what are the computational and data requirements (for both training and inference) that constitute the practical threshold for deployment?
>
> Our current setup operates in a low-data regime, using about 200 trajectories per object (\~6 minutes of footage; see Dataset section), and is computationally lightweight for both training (~6 days on one H100) and inference (16 FPS (4 steps) and 20.1 FPS (2 steps), under 20 GB VRAM on an H100)
>
> The model was designed to scale naturally to larger datasets by avoiding hand-engineered features and privileged information, which we believe makes foundation-style training feasible in future work. As noted in the Discussion, two main challenges remain before scaling to large-scale data: (i) incorporating action-conditioning to enable causal world modeling, as in 3DGSim, and/or (ii) developing a learnable mechanism that allows the model to dynamically create or remove particles and represent 3D dynamics in a generative manner, analogous to recent advances in video generation.

---

### Author Response · Authors · 2025-12-03
**Summary of Rebuttal Updates and Key Contributions**

Dear Area Chairs and Program Chairs,

In this final comment, we summarize how we have addressed the reviewers' concerns and highlight the new results added during the discussion period.

We engaged actively with the reviewers, providing: \
**(i) New Experiments:** Quantitative contact timing analysis (Reviewer X9bZ) \
**(ii) Comprehensive Ablations:** Detailed breakdown of key components (Reviewers jav5, KeGc) \
**(iii) Clarified Positioning:** Comparison with parameter-recovery baselines and discussion of datasets (Reviewers X9bZ, qMN5)

We received positive feedback from Reviewer X9bZ (25 Nov), who acknowledged the adequacy of our rebuttal and raised two follow-up points regarding contact timing and comparisons with Vid2Sim, both of which we have addressed below. While we could not further engage with the other reviewers, we believe the additional data provided in our rebuttal fully answers their initial questions.


**1. Motivation & novelty** (Overall Clarification)

Our contribution is a framework that unifies 3D reconstruction and generative physics, allowing for learned spatial causal world modeling and offering a novel direction for learning simulators directly from visual data.

To demonstrate the necessity of this 3D inductive bias, we compare 3DGSim against a fine-tuned version of Cosmos (SOTA video generation). In out-of-distribution scenarios (such as removing the ground or introducing multi-object collisions despite only training on single-body scenes), 3DGSim maintains physical consistency. In contrast, 2D baselines exhibit catastrophic failures (e.g., objects hovering or merging), demonstrating that a 3D formulation is essential for robust generalization.

**2. Comprehensive Ablation Study** (Addressing jav5, KeGc)

In response to the request, we consolidated the ablation results, previously distributed across multiple appendix tables, into a single summary (Figure 14 in the revised paper). Conducted on the elastic dataset, these experiments demonstrate the impact of our specific design decisions.

| **Variant**                            | **PSNR Future**               | **Impact Analysis**                                          |
| ------------------------------- | ------------------------------------ | ------------------------------------------- |
| **Final Model (Latent 4-1-12)** |  **33.15 ± 3.51**             | **Best performance.**                                        |
| Input Steps (2-1-12)            |  32.05 ± 3.48                 | Slight drop with less context.                               |
| Modality (Predict All 4-4-12)   | 31.98 ± 3.78                 | Predicting all steps degrades stability.                     |
| Feature Rep. (Explicit 3DGS)    | 29.69 ± 1.75                 | **Latent > Explicit.**  \*\* |
| Rollout Length (4-1-2)          | 26.43 ± 3.48                 | Long-horizon training (12 steps) is crucial for stability.   |
| **Without TEM**                 | **18.19 ± 1.29**             | **TEM is Critical.** Removing Temporal Encoding/Merging causes collapse. |

*\*\* Abstract features allow embedding physical properties better than explicit visual features.*

**3. Contact Timing Analysis** (Addressing X9bZ)

To quantitatively assess physical fidelity beyond visual metrics, we implemented a Time of Impact (ToI) experiment using 10 generated trajectories per object with privileged simulator states. We measured the absolute frame error between the predicted contact and ground truth, utilizing a KNN-based filtering step to remove artifacts (e.g., semi-transparent floaters).

Results (Absolute Frame Error):
- Elastic: 0.69 \pm 0.72 frames
- Cloth: 1.10 \pm 1.13 frames
- Rigid: 1.36 \pm 1.11 frames

**Conclusion:** 3DGSim predicts impact timing with an average error of ~1 frame. This confirms that the model learns accurate implicit physics, not just visual interpolation.

**4. Comparisons & Baselines** (Addressing X9bZ, qMN5)

We clarified the distinction between parameter-recovery approaches (e.g., NeuMa, PAC-NeRF, Vid2Sim, PhysGaussian) and our fully data-driven approach.
- **Baselines:** Typically rely on black-box simulators with known constitutive models, requiring per-scene parameter recovery and manual boundary conditions (e.g., manually adding the floor).
- **3DGSim:** Learns dynamics end-to-end from video without physical priors or manual boundary conditions.

Regarding PhysGaussian datasets (suggested by Reviewer qMN5), we clarified that these are static reconstruction datasets which they only post-simulated for evaluation. As they lack ground-truth dynamic interactions, they are unsuitable for benchmarking learned dynamics.

We thank you for taking the time to review this rebuttal.

Best regards,
The Authors

---

### Meta-Review · Area_Chair_Hff2 · 2026-01-04

**Summary:**

The paper proposes a differentiable framework that learns 3D particle based simulators from multi-view RGB videos. Different from prior work which explicitly recover the physical params, it learns the dynamics from videos directly in an implicit end-to-end fashion. It builts upon MVSplat, a transformer-based dynamic modeling block, and a 3D-GS renderer for differentiable image supervision.

The paper initially received four reviews, i.e. 1 positive rating (rating 6) and 3 negative ratings (rating 4). The main concerns regarding the paper are 1) lack of validation on real data (current evaluations are mainly conducted with simple unrealistic synthetic data); 2) lack of quantitative and qualitative comparisons with relevant 3D baselines. The AC agrees with the reviewers that it would be difficult to adequately demonstrate the advantages of the proposed method with missing real-data evaluations. Thus, the AC encourages the authors to address the above concerns and submit it to anoter venue.

**Reviewer Concerns:**

1. All reviewers raised the same concern regarding the lack of evaluations on real data, the rebuttal does not address this critical concern.
2. Another concern on lack of comparisions against prior 3D baselines, raised by X9bZ, qMN5 and jav5, is also not being well addressed.

**Reviewer Scores:**

Since the rebuttal does not fully address the critical concerns from the reviewers, the AC think the reviewers would not upgrade their ratings.

---

### Decision · Program_Chairs · 2026-01-26

Reject